



Atmospheric
Chemistry
and Physics

# Simulation of organic aerosol formation during the CalNex study: updated mobile emissions and secondary organic aerosol parameterization for intermediate-volatility organic compounds

Quanyang Lu[1,2,3], Benjamin N. Murphy[4], Momei Qin[3,a], Peter J. Adams[1], Yunliang Zhao[1,2,b], Havala O. T. Pye[4], Christos Efstathiou[5], Chris Allen[5], and Allen L. Robinson[1,2]

[1]Center of Atmospheric Particle Studies, Carnegie Mellon University, Pittsburgh, PA, USA
[2]Department of Mechanical Engineering, Carnegie Mellon University, Pittsburgh, PA, USA
[3]Oak Ridge Institute for Science and Education (ORISE) Research Participation Program at the Office of Research and Development, U.S. Environmental Protection Agency, Research Triangle Park, NC, USA
[4]Office of Research and Development, U.S. Environmental Protection Agency, Research Triangle Park, NC, USA
[5]General Dynamics Information Technology Research Triangle Park, North Carolina, USA
[a]now at: Nanjing University of Information Science and Technology, Nanjing, China
[b]now at: California Air Resources Board, Sacramento, CA, USA

**Correspondence:** Allen Robinson (alr@andrew.cmu.edu) and Benjamin Murphy (murphy.benjamin@epa.gov)

**Abstract.** TS1 We describe simulations using an updated version of the Community Multiscale Air Quality model version 5.3 (CMAQ v5.3) to investigate the contribution of intermediate-volatility organic compounds (IVOCs) to secondary organic aerosol (SOA) formation in southern California during the CalNex study. We first derive a model-ready parameterization for SOA formation from IVOC emissions from mobile sources. To account for SOA formation from both diesel and gasoline sources, the parameterization has six lumped precursor species that resolve both volatility and molecular structure (aromatic versus aliphatic). We also implement new mobile-source emission profiles that quantify all IVOCs based on direct measurements. The profiles have been released in SPECIATE 5.0. By incorporating both comprehensive mobile-source emission profiles for semivolatile organic compounds (SVOCs) and IVOCs and experimentally constrained SOA yields, this CMAQ configuration best represents the contribution of mobile sources to urban and regional ambient organic aerosol (OA). In the Los Angeles region, gasoline sources emit 4 times more non-methane organic gases (NMOGs) than diesel sources, but diesel emits roughly 3 times more IVOCs on an absolute basis. The revised model predicts all mobile sources (including on- and off-road gasoline, aircraft, and on- and off-road diesel) contribute $\sim 1\,\mu g\,m^{-3}$ to the daily peak SOA concentration in Pasadena. This represents a $\sim 70\,\%$ increase in predicted daily peak SOA formation compared to the base version of CMAQ. Therefore, IVOCs in mobile-source emissions contribute almost as much SOA as traditional precursors such as single-ring aromatics. However, accounting for these emissions in CMAQ does not reproduce measurements of either ambient SOA or IVOCs. To investigate the potential contribution of other IVOC sources, we performed two exploratory simulations with varying amounts of IVOC emissions from nonmobile sources. To close the mass balance of primary hydrocarbon IVOCs, IVOCs would need to account for 12 % of NMOG emissions from nonmobile sources (or equivalently $30.7\,t\,d^{-1}$ in the Los Angeles–Pasadena region), a value that is well within the reported range of IVOC content from volatile chemical products. To close the SOA mass balance and also explain the mildly oxygenated IVOCs in Pasadena, an additional 14.8 % of nonmobile-source NMOG emissions would need to be IVOCs (assuming SOA yields from the mobile IVOCs apply to nonmobile IVOCs). How-

Please note the remarks at the end of the manuscript.

ever, an IVOC-to-NMOG ratio of 26.8 % (or equivalently 68.5 t d$^{-1}$ in the Los Angeles–Pasadena region) for nonmobile sources is likely unrealistically high. Our results highlight the important contribution of IVOCs to SOA production in the Los Angeles region but underscore that other uncertainties must be addressed (multigenerational aging, aqueous chemistry and vapor wall losses) to close the SOA mass balance. This research also highlights the effectiveness of regulations to reduce mobile-source emissions, which have in turn increased the relative importance of other sources, such as volatile chemical products.

## 1 Introduction

Exposure to fine particulate matter (PM$_{2.5}$ and PM$_1$) has been associated with increased mortality, lung cancer and cardiovascular diseases (Apte et al., 2018; Di et al., 2017). Organic aerosol (OA) is a major component of ambient fine particulate matter (Jimenez et al., 2009; Zhang et al., 2015). The majority of OA, even in most urban areas, is secondary organic aerosol (SOA), formed from the atmospheric oxidation of gas-phase species. Over the past several decades, primary emissions have been greatly reduced in the United States, which has led to significant improvement in air quality, especially in the Los Angeles Basin in California (Warneke et al., 2012; Zhang et al., 2018). However, SOA remains an important component of fine particulate matter, but its sources are uncertain (Ensberg et al., 2014; McDonald et al., 2018).

Intermediate-volatility organic compounds (IVOCs) are an important class of SOA precursors (Chan et al., 2009; Liggio et al., 2016; Presto et al., 2009; Zhao et al., 2014). IVOCs, for example, C$_{12}$ to C$_{17}$ $n$-alkanes and polycyclic aromatic hydrocarbons, are efficient SOA precursors (Chan et al., 2009; Presto et al., 2010a). In addition, chamber experiments using unburned fuel and diluted exhaust have demonstrated the importance of IVOCs to SOA production from mobile-source emissions (Gordon et al., 2014; Jathar et al., 2013; Miracolo et al., 2011; Platt et al., 2017).

Despite this evidence, IVOCs are not routinely or consistently accounted for in chemical transport models (CTMs). A major challenge has been the lack of emissions data due to a combination of sampling challenges and the fact that the vast majority of IVOC emissions have not been speciated on a molecular basis. In addition, chemical mechanisms (e.g., SAPRC and carbon bond) often do not explicitly account for IVOCs, instead lumping them with VOCs or nonreactive gases (Lu et al., 2018). Several recent studies report total (speciated and unspeciated) IVOC emissions from a variety of mobile sources, including on- and off-road gasoline, diesel, aircraft, and vessel engines (Cross et al., 2013; Huang et al., 2018; Kroll et al., 2014; Pereira et al., 2018; Presto et al., 2011; Qi et al., 2019; Wang et al., 2012; Zhao et al., 2015,

2016). While these studies have not been able to speciate all of the IVOCs emissions at the molecular level, some provide insight into the molecular structure of the unspeciated IVOCs (Drozd et al., 2019; Hatch et al., 2017; Hunter et al., 2017; Worton et al., 2014; Zhao et al., 2015, 2016). For example, IVOCs in diesel exhaust are primarily comprised of aliphatic compounds, while IVOCs in gasoline exhaust are primarily aromatics with higher OH reaction rates and SOA yields. Zhao et al. (2015, 2017) used these new emissions data to explain the SOA formation in smog chamber experiments with diluted vehicle emissions. The SOA mechanism proposed by Zhao et al. (2015, 2017) accounts for all of the IVOC emissions. It represents them using 79 different compounds, some of which are individual species and others lumped groups assigned based on gas chromatography and mass spectrometry data. However, this model is too computationally expensive for implementation in current operational CTMs.

Because of the high levels of both ozone and PM exposure in the Los Angeles Basin over the last several decades, extensive ambient measurement campaigns have explored the sources of poor air quality in the region, including the CalNex campaign in 2010 (Ryerson et al., 2013). During the CalNex campaign, average OA at the Pasadena supersite was 7 µg m$^{-3}$, of which SOA, defined as the sum of semivolatile and low-volatility oxygenated OA (SV-OOA and LV-OOA) factors from AMS analysis, contributed 66 % to total OA mass (Hayes et al., 2013). Zhao et al. (2014) measured the ambient IVOC concentration at the Pasadena site and estimated that photooxidation of IVOCs contributed up to 57 % of SV-OOA during CalNex.

A number of chemical transport model (CTM) studies have examined SOA formation in the Los Angeles Basin during the CalNex campaign (Baker et al., 2015; Fast et al., 2014; Jathar et al., 2017; Murphy et al., 2017; Woody et al., 2016). However, these studies used very different assumptions for IVOC emissions and their SOA yields. IVOC emissions are commonly estimated by applying a scaling factor to some other species (generally primary organic aerosol, POA). These scaling factors have been based on little experimental data, and typically the same factor is applied to all sources. For example, Fast et al. (2014) assumed additional SOA precursor (IVOC and/or SVOC, semivolatile organic compound) mass of 6.5× POA and Woody et al. (2016) assumed 7.5× POA based on previous estimations (Hodzic et al., 2010; Koo et al., 2014), applied to all emission source categories. Jathar et al. (2017) assumed mobile IVOC emission as 25 % of diesel non-methane organic gas (NMOG) emissions and 20 % of gasoline NMOG emissions. Finally, Baker et al. (2015) did not explicitly account for IVOCs but increased the SOA yields from VOCs by a factor of 4 compared to the base version of the Community Multiscale Air Quality (CMAQ) model. Despite these efforts, these studies still underpredicted the measured OA by a factor of 2 to 6 (Hayes et al., 2013). Murphy et al. (2017) largely closed the OA mass balance by defining a new lumped SOA precur-

sor called potential combustion volatile organic compounds (pcVOCs) with emissions equal to $9.6\times$ POA and an SOA yield of 1. However, all the abovementioned models used scaling factors that are not based on actual emission data. They also only use a single IVOC surrogate, which does not account for differences in IVOC chemical composition. Lu et al. (2018) showed that a single scaling factor does not represent the magnitude of actual IVOC emissions across all mobile sources. Finally, none of these models account for the effects of differences in molecular structure in IVOC emissions on SOA yield.

Mobile sources are major sources of NMOG emissions and therefore important sources of SOA precursors in urban environments (Gentner et al., 2017). Historically, mobile sources have been the dominant source of NMOGs in many urban areas, but their contribution has been reduced due to increasingly stringent emission regulations. The 2014 EPA National Emission Inventory (NEI) estimates that mobile sources contribute 32 % of the anthropogenic VOC emissions nationally (and 43 % in Los Angeles County). In Los Angeles County, on- and off-road gasoline and diesel sources account for more than 96 % of mobile-source emissions.

Lu et al. (2018) recently compiled mobile-source emission data, including on- and off-road gasoline, aircraft and diesel engines, to create updated model-ready emission profiles that include explicit treatment of IVOCs. They found that mobile-source NMOG emissions can be explained by trimodal distributions of by-product, fuel and oil modes. IVOC emissions originate from fuel components, and similar distributions are observed across sources that use the same fuel (Cross et al., 2015; Lu et al., 2018; Presto et al., 2011). This applies to both low-emitting heavily controlled sources (e.g., LEV-II-certified gasoline vehicle) and uncontrolled high-emitting sources (e.g., two-stroke gasoline off-road sources) (Lu et al., 2018). Therefore, in this work, mobile IVOC emissions are modeled and grouped based on fuel type.

In this paper, we use an updated version of CMAQ v5.3 (US EPA Office of Research and Development, 2019) to investigate the sources and contribution of SVOCs and IVOCs to SOA formation in the Los Angeles region during the CalNex campaign. We updated CMAQ v5.3 with a new set of mobile-source NMOG and SVOC emission profiles that include six classes of IVOCs and a new parameterization of SOA formation from IVOC precursors designed for implementation into chemical transport models. The new emission profiles are based on direct measurement of IVOCs from on- and off-road mobile sources (Gordon et al., 2013; Lu et al., 2018; May et al., 2014; Presto et al., 2011; Zhao et al., 2015, 2016). These profiles (100VBS to 103VBS) are now available in SPECATE 5.0 (US EPA, 2019). The new SOA parameterization is derived from a comprehensive parameterization that explains the SOA formation from dilute mobile-source exhaust in smog chamber studies (Zhao et al., 2015, 2017). We evaluate the resulting model, now the most-up-to-date representation of mobile-source organic compound emissions, using data collected during the CalNex campaign, including direct measurements of ambient IVOCs. Finally, we explore the potential contribution of nonmobile sources to IVOC and OA concentrations.

## 2 Parameterizing SOA formation from mobile-source IVOCs

Mobile sources are comprised of a complex mixture of on- and off-road sources, including gasoline, aircraft and diesel engines. However, they are predominantly gasoline- and diesel-powered, with a small fraction of aircraft emissions. In this work we apply the source profiles of Lu et al. (2018) to estimate the amount and composition of the IVOC emissions for different mobile sources. The IVOCs are normalized to total NMOG emissions, which only include the organics in the volatility range from $C^* = 10^3$ to $10^{11}\,\mu g\,m^{-3}$. Table 1 summarizes the IVOC-to-NMOG ratios for different mobile sources. The ratios (and associated emission profiles) vary widely depending on the underlying fuel. For gasoline, aircraft and diesel sources, IVOCs comprise 4.6 %, 28.5 % and 55.5 % of the NMOG emissions, respectively. IVOC emissions from gasoline sources include high fractions of aromatics (Drozd et al., 2019; Zhao et al., 2016).

We developed a simplified parameterization to simulate first-generation SOA formation from IVOCs under high-$NO_x$ conditions. By first generation we mean the amount of SOA that forms within a couple of hours in a smog chamber experiment with dilute exhaust at typical atmospheric oxidant levels. The parameterization is derived from the model of Zhao et al. (2015, 2016), which explicitly accounts for 79 different classes of IVOCs. The chemistry and transport associated with 79 additional species in the gas and particle phases would be too computationally expensive in a CTM which normally has about 50 or fewer organic aerosol species. Our aim is to develop a model for IVOC SOA production that can be used in off-the-shelf regulatory and routine chemical transport modeling applications. For other applications, a more-explicit approach with multiple thousands of species may be more powerful for modeling reaction pathways (Ying and Li, 2011). From the IVOC measurement perspective, lumping similar IVOCs together based on their volatility and functionality is also more interpretable and compatible with data provided by most instruments.

The Zhao et al. (2015, 2016) model accounts for 57 individual IVOCs and 22 lumped IVOCs. The 22 lumped IVOCs are comprised of unspeciated IVOCs grouped based on gas chromatography (GC) retention time and an assigned chemical class based on its mass spectra. This model explains the SOA formation from dilute exhaust of gasoline and diesel vehicles measured in chamber experiments (Zhao et al., 2015, 2017). Our simplified SOA parameterization accounts for the key differences in chemical composition of the IVOC emissions from different mobile sources. This is important

**Table 1.** Mass fractions (grams per gram of NMOG) of IVOCs in mobile NMOG emission profiles used in CMAQ simulations.

| Group | | Volatility | Source | | |
|---|---|---|---|---|---|
| | | ($C^*$ at 298 K, µg m$^{-3}$) | Gasoline | Aircraft | Diesel |
| Non-aromatics | IVOCP6-ALK | $10^6$ | 0.006 | 0.207 | 0.159 |
| | IVOCP5-ALK | $10^5$ | 0.002 | 0.048 | 0.187 |
| | IVOCP4-ALK | $10^4$ | 0.003 | 0.020 | 0.149 |
| | IVOCP3-ALK | $10^3$ | 0.003 | 0.009 | 0.054 |
| Aromatics | IVOCP6-ARO | $10^6$ | 0.025 | NA | 0.002 |
| | IVOCP5-ARO | $10^5$ | 0.006 | NA | 0.004 |
| Total | | | 0.046 | 0.285 | 0.555 |

NA – not available

because the composition of the IVOC emissions varies by source class (e.g., gasoline versus diesel), and SOA yield depends on both molecular weight (volatility) and chemical structure (aromatics versus alkanes) (Chan et al., 2009; Jathar et al., 2013; Lim and Ziemann, 2005, 2009; Presto et al., 2010a). For example, diesels emit more lower-volatility IVOCs than gasoline engines, but diesel IVOC emissions are mainly comprised of aliphatic compounds versus aromatics for gasoline. These differences matter because, for a given chemical class, SOA yields generally increase with increasing molecular weight, which increases the effective SOA yield of diesel exhaust relative to gasoline exhaust. However, for a given carbon number, the SOA yield for hydrocarbon IVOCs generally follows aromatics > cyclic > linear > branched alkanes (Lim and Ziemann, 2009; Tkacik et al., 2012); thus gasoline IVOC yields increase when their structure is considered. Finally, aromatic IVOCs have higher OH reaction rates than alkanes (Chan et al., 2009; Zhao et al., 2017). In this study, we only account for IVOC–OH reactions because mobile-source IVOCs are mostly alkanes or aromatics, which will react slower with $O_3$. $NO_3$ oxidation can be important in nighttime SOA formation (Fry et al., 2014; Hoyle et al., 2011), and this will be important to consider in the future, but experimental studies on SOA formation from anthropogenic IVOC reactions with $NO_3$ radical are limited at this time.

To illustrate the complexity of the IVOC mechanisms of Zhao et al. (2015, 2016), Fig. 1 plots the SOA yield (expressed as SOA mass divided by mass of precursor) as a function of volatility for the 79 different IVOCs in the model at a typical atmospheric OA concentration of 10 µg m$^{-3}$. This model likely provides a conservative estimate for SOA yields of lower-volatility IVOCs, as C$_{18-22}$ $n$-alkanes are assumed to have the same SOA yields as C$_{17}$ $n$-alkanes. The scatter in the data highlights the complex relationship between molecular structure and SOA yield.

Our goal is to derive a semi-empirical SOA parameterization with the minimum number of surrogate species that

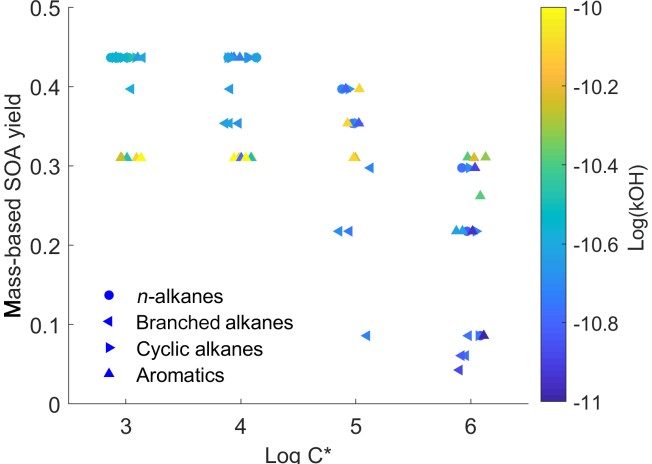

**Figure 1.** CE1 Scatter plot of first-generation mass-based SOA yields versus volatility (log $C^*$, µg m$^{-3}$) in the detailed parameterization (dots are colored by OH reaction rates).

reproduces the mechanism of Zhao et al. (2015, 2016). The simplified parameterization must account for the differences in SOA formation from IVOC emissions from different mobile-source categories (gasoline, diesel and aircraft). We developed the simplified parameterization using the volatility basis set (VBS) framework of Donahue et al. (2006) following the approach of Presto et al. (2010b). The parameterization accounts for all IVOC emissions, which are lumped into surrogates based on gas chromatography retention time (related to volatility) and mass spectral (composition information) data (Lu et al., 2018). Like the work of Zhao et al. (2015, 2016), the parameterization accounts for all IVOC mass, not just the mass that can be speciated at the molecular level (Lu et al., 2018). Briefly, to simulate SOA formation, each lumped IVOC group reacts with OH to form a set of semivolatile products in Eq. (1):

$$\text{IVOC}_i + \text{OH} \rightarrow \alpha_{i,1} P_{C^*=0.1} + \alpha_{i,2} P_{C^*=1} + \alpha_{i,3} P_{C^*=10}$$
$$+ \alpha_{i,4} P_{C^*=100} \text{ for group } i = 1 \text{ to } 6, \quad (1)$$

where $\alpha_{i,1}$ to $\alpha_{i,4}$ are mass-based stoichiometric coefficients for $IVOC_i$ distributing the reaction products across a second volatility basis set from 0.1 to 100 µg m$^{-3}$ (Presto et al., 2010b). For each lumped IVOC species there are five unknowns: four stoichiometric coefficients ($\alpha_{i,1}$ to $\alpha_{i,4}$) and the OH reaction rate $k_{OH,i}$. These coefficients and reaction rates are derived by fitting the mechanism of Zhao et al. (2015, 2016). All SOA parameters are set at a fixed temperature of 298 K. Details of the fitting procedure are in the Supplement.

We initially tried using four lumped-IVOC-species distributed across the volatility basis set ($C^* = 10^3$ to $10^6$ µg m$^{-3}$) to account for the influence of precursor volatility based on gas chromatography retention time but not molecular structure on SOA yield. However, that model poorly reproduced the SOA formation from gasoline vehicle emissions, especially at shorter timescales (Fig. S1 in the Supplement). The problem is that IVOCs in diesel exhaust are dominated by aliphatic compounds, while IVOCs in gasoline exhaust are dominated by aromatics (Drozd et al., 2019; Zhao et al., 2016); as previously discussed, aromatic compounds have different OH reaction rates and SOA yields (Fig. 1) (Lim and Ziemann, 2009; Tkacik et al., 2012).

We therefore defined two additional lumped IVOC species with $C^* = 10^5$ and $10^6$ µg m$^{-3}$ to account for the aromatic IVOCs in gasoline engine exhaust (Table 1). The IVOCs in these two bins were split based on mass spectral data (Zhao et al., 2015, 2016). Mobile-source IVOC emissions in the lower-volatility bins of $C^* = 10^3$ and $10^4$ µg m$^{-3}$ are primarily alkanes from unburned fuel or lubricant oil (Lu et al., 2018; Worton et al., 2014); therefore, the simplified mechanism only includes one lumped aliphatic IVOC species in each of those bins. IVOC emissions are assigned to these surrogate species using the source profiles listed in Table 1.

To illustrate the performance of the new parameterization, Fig. 2a compares the predicted SOA using our six-IVOC-group parameterization to the original mechanism of Zhao et al. (2015, 2016). It shows that the two models agree with an absolute error for the mass-based SOA yield of less than 0.01 for all mobile sources at an OA concentration of 5 µg m$^{-3}$. Across a wide range of atmospherically relevant concentrations (OA of 1 to 50 µg m$^{-3}$), Figure 2(b) shows that the relative error is less than 6 % between our new parameterization and the original mechanisms of Zhao et al. (2015, 2016).

The yields derived by the fitting make physical sense. The yields increase with decreasing volatility (Table 2). The fitting procedure assigns higher yields and faster reaction to the lumped aromatics compared to aliphatics in the same volatility bin (Drozd et al., 2019; Zhao et al., 2016). This explains the higher SOA production in the first 10 h from gasoline exhaust compared to aircraft and diesel IVOC emissions. It also predicts that diesel IVOC emissions have the overall highest SOA yield due to their high fraction of lower-volatility compounds compared to emission from gasoline engines and aircraft (Lu et al., 2018; Zhao et al., 2015).

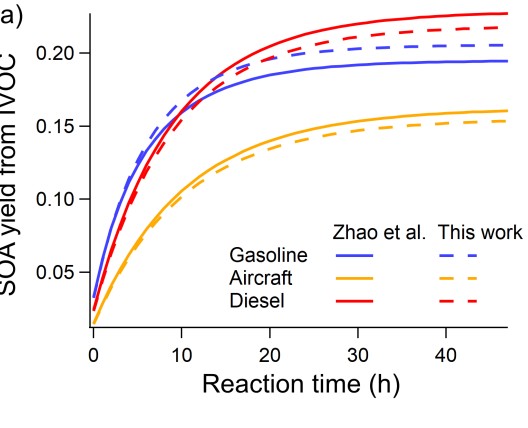

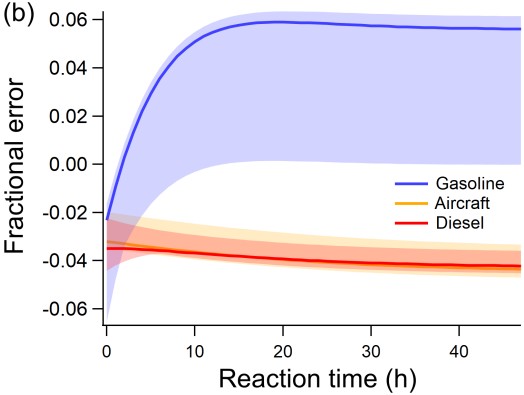

**Figure 2. (a)** Comparison of predicted SOA formation per unit of mass mobile IVOC emissions of new parameterizations and model of Zhao et al. (2015, 2016) at OA $= 5$ µg m$^{-3}$ (average [OH] $= 3 \times 10^6$ cm$^{-3}$). **(b)** Relative error in SOA formed between new and Zhao et al. (2015, 2016) parameterizations (solid line is the relative error at OA $= 5$ µg m$^{-3}$; shaded area corresponds to OA from 1 to 50 µg m$^{-3}$).

Table 2 lists the set of $k_{OH}$ and $\alpha_i$ for the simplified six-IVOC-group parameterization for mobile-source emissions. Molecular weights (MWs) are determined as the average MWs of $n$-alkanes or speciated aromatics in each volatility bin. The IVOC MWs are used to convert mass-based SOA yields to molar units and calculate parameters needed to simulate dry deposition processes. Enthalpies of vaporization ($H_{vap}$) are determined using the fitted parameterization in Ranjan et al. (2012). In this work, we implement this six-lumped-IVOC-group parameterization to model the IVOC SOA formation in CMAQ v5.3. The first-generation products represented in Eq. (1) undergo multigenerational aging following the mechanism of Murphy et al. (2017) described in Sect. 3.4.

## 3 CMAQ model

To evaluate the contribution of mobile-source IVOC emissions to ambient SOA, we implemented our new six-lumped-

**Table 2.** Properties and stoichiometric mass-based product yields for six-group IVOC–SOA parameterization.

| Group | $C^*$ ($\mu g\,m^{-3}$, at 298 K) | MW ($g\,mol^{-1}$) | $k_{OH} \times 10^{11}$ ($cm^{-1}\,molec^{-1}\,s^{-1}$) | $\alpha_i$ ($C^* = 0.1$ to $100\,\mu g\,m^{-3}$ at 298 K) | | | | Yield at $10\,\mu g\,m^{-3}$ | $H_{vap}$ ($kJ\,mol^{-1}$) |
|---|---|---|---|---|---|---|---|---|---|
| | | | | 0.1 | 1 | 10 | 100 | | |
| IVOCP6-ALK | $10^6$ | 184.4 | 1.55 | 0.009 | 0.045 | 0.118 | 0.470 | 0.15 | 19 |
| IVOCP5-ALK | $10^5$ | 219.4 | 1.89 | 0.051 | 0.061 | 0.394 | 0.494 | 0.35 | 30 |
| IVOCP4-ALK | $10^4$ | 254.9 | 2.25 | 0.068 | 0.083 | 0.523 | 0.239 | 0.43 | 41 |
| IVOCP3-ALK | $10^3$ | 296.6 | 2.65 | 0.067 | 0.086 | 0.544 | 0.198 | 0.43 | 52 |
| IVOCP6-ARO | $10^6$ | 162.3 | 3.05 | 0.022 | 0.109 | 0.251 | 0.005 | 0.25 | 19 |
| IVOCP5-ARO | $10^5$ | 197.3 | 7.56 | 0.143 | 0.021 | 0.329 | 0.358 | 0.36 | 30 |

IVOC-group SOA parameterization and emission profiles into CMAQ v5.3. We used the model to simulate the air quality in California from 1 May to 30 June 2010, which includes the entire CalNex campaign (May and July 2010). Except as noted below, the simulations described here have essentially the same modeling domain and input parameters as previous modeling studies on CalNex (Baker et al., 2015; Murphy et al., 2017; Woody et al., 2016). We have extended this previous work by updating the emissions and SOA formation from IVOCs.

## 3.1 Model configuration

The model domain covered California and Nevada with a 4 km ($325 \times 225$) grid resolution and 35 vertical layers. The input meteorology and NEI emission inventory are very similar to those used by Baker et al. (2015), Woody et al. (2016), and Murphy et al. (2017) and are identical to Qin et al. (2020). Meteorological inputs were generated using the Weather Research and Forecasting Model (WRF) Advanced Research WRF core version 3.8.1 (Skamarock et al., 2008) with one additional model layer at the surface compared to previous studies (i.e., the lowest layer of approximately 40 m depth has been split into two 20 m deep layers to better resolve surface gradients). The emission inputs are based on the 2011 NEI version 2 with mobile, wildfire and electric-generating point source emissions calculated for 2010. Mobile on-road and off-road emissions are calculated by MOVES 2014a, except that on-road emissions for California are estimated by EMFAC and allocated using MOVES 2014a. Biogenic emissions are calculated online with BEIS v3.61 and improved land use cover from BELD4 (Bash et al., 2016). Sea-spray aerosols are calculated online and incorporate dynamic prediction of particle population size and standard deviation. Windblown dust emissions are neglected and should not impact comparisons with the data collected by the AMS, which detects non-refractory particulate compounds. Moreover, previous studies (Cazorla et al., 2013) found little evidence of dust impacts during CalNex using both in situ aircraft measurements and inference from AERONET retrievals. Gas-phase chemistry is simulated with the SAPRC07T chemical mechanism (Carter, 2010; Hutzell

et al., 2012; Xie et al., 2013). Aerosols are simulated using the Aero7 CE2 module (CMAQ-AE7) with monoterpene photooxidation updates (Xu et al., 2018) and organic water uptake (Pye et al., 2017). Boundary conditions were generated from a 12 km continental United States simulation of April to June 2010. We use the first 14 d CE3 of the simulation as a spin-up to minimize the influence of initial conditions.

Previous studies (Baker et al., 2015; Woody et al., 2016) have extensively evaluated different versions of CMAQ using CalNex data. These evaluations show good to excellent performance for many pollutants, with a notable exception of organic aerosols and SOA – the focus of this paper. We evaluated our model predictions with measurements of gas-phase pollutants such as CO, $O_3$ and $NO_x$ as they are typical indicators for model performance. Consistent with the previous applications of CMAQ to CalNex (Baker et al., 2015; Murphy et al., 2017; Woody et al., 2016), Fig. S2a shows very good agreement between modeled and measured CO diurnal patterns in Pasadena, and the normalized mean bias (NMB) is 4.2 %. Figure S3 compares the $O_3$, NO and $NO_2$ diurnal patterns with measurements in Pasadena, where the NMB is 10.7 %, −6.7 % and 5.4 %, respectively. Figure S4 compares the CO, $O_3$ and NO diurnal patterns for three other sites: Bakersfield, Sacramento and Cool. The model NMB is within ±25 % for all comparisons, except for $O_3$ and NO in Bakersfield. Thus, we can conclude that the CMAQ model performs reasonably well at all four sites for traditional gas-phase pollutants.

## 3.2 POA emissions

CMAQ v5.3 treats POA emissions as semivolatile with variable gas–particle partitioning and multigenerational aging (Fig. S5). The POA model, similar to the 1.5-dimensional VBS of Koo et al. (2014), contains five pairs of hydrocarbon-like vapor/particle species (one LVOC, three SVOCs and one IVOC) distributed across a volatility basis set with $C^*$ from $10^{-1}$ to $10^3\,\mu g\,m^{-3}$, with O : C increasing slightly with decreasing volatility. POA emissions are then assigned to each of these species using the source-specific volatility profiles in Table 3, and CMAQ calculates gas–particle partitioning assuming equilibrium partitioning and treating the entire or-

**Table 3.** POA volatility distributions and filter artifact scaling factors.

| Source | Volatility, $C^*$ ($\mu g\,m^{-3}$, at 298 K) | | | | Filter artifact scaling factor |
| --- | --- | --- | --- | --- | --- |
| | $\leq 10^{-1}$ | 1 | 10 | $10^2$ | |
| Gasoline | 0.16 | 0.08 | 0.37 | 0.39 | 1.4 |
| Diesel | 0.21 | 0.11 | 0.33 | 0.36 | 1 |
| Gas turbine | 0.15 | 0.26 | 0.38 | 0.21 | 1 |

ganic phase as a single, pseudo-ideal solution. For nonmobile sources, POA emissions are distributed into all five bins with $C^*$ from $10^{-1}$ to $10^3\,\mu g\,m^{-3}$, while the mobile-source POA profiles only map to the $10^{-1}$ to $10^2\,\mu g\,m^{-3}$ bins.

Comprehensive emission profiles for semivolatile POA include both SVOCs and lower-volatility organics (Lu et al., 2018). In the base version of CMAQ v5.3, the volatility profile of Robinson et al. (2007) is used to represent all combustion sources. Here, we update the volatility distributions for mobile POA using the new mobile-source emission profiles in Lu et al. (2018). The profiles (8873VBS and 8992VBS to 8996VBS) are available in SPECIATE 5.0 (US EPA, 2019). For nonmobile combustion sources, we use the biomass-burning POA volatility distribution from May et al. (2013b) for wood-burning sources, the cooking POA volatility distribution from Woody et al. (2016) for cooking sources and the diesel POA volatility distribution from May et al. (2013a) as a surrogate for all other combustion sources. According to our emission inventory, mobile, wood-burning and cooking sources combined emit more than 80 % of total POA in the Los Angeles region during the modeled period, where other combustion sources only emit 16.4 % of the POA. We acknowledge that the diesel POA surrogate is modestly more volatile than biomass-burning POA profiles. Thus, using diesel POA volatility as the surrogate for other combustion sources will possibly increase the regional SOA formation compared to if a different profile was used, but the potential bias is small. Table 3 summarizes the volatility distributions and scaling factors used in this work. The same POA emissions were used for all model runs.

A challenge is that most existing POA emission factors used to inform inventories such as NEI are based on filter measurements, which do not quantitatively collect all SVOCs. For example, filters collect only a portion of SVOC vapors. Estimating this error is complex because there are competing biases. First, source testing is often performed at low levels of dilution, which creates high concentrations (relative to the more dilute atmosphere) that shift gas–particle partitioning of SVOCs to the particle phase. In these situations, filters collect a larger fraction of SVOCs than more dilute conditions (of course, at high enough concentrations, filters will also collect some IVOC vapors). Second, during mobile-source testing, filters are commonly collected at elevated temperatures (e.g., 47 °C) to avoid water condensation,

which shifts gas–particle partitioning towards the gas phase, reducing the fraction of SVOCs collected by a filter. Finally, filters collect some vapors as sampling artifacts, which depends on many factors, including filter material, filter face velocity and filter pretreatment (Subramanian et al., 2004). Therefore, the fraction of SVOCs collected by filters depends on these competing effects, which are difficult to quantify. As expected, data from Zhao et al. (2015, 2016) and Lu et al. (2018) indicate that the fraction of SVOCs collected depends on the OA concentration inside the sampling system.

To estimate potential biases in the amount of SVOC vapors in the filter-based POA emission factor measurements, we compared the mass of lower-volatility organics (SVOC + LVOC + NV) collected on filters and Tenax tubes versus the mass collected on filters (regular POA measurement) (Lu et al., 2018). The two estimates for diesel and gas-turbine tests were within 10 %, which is within experimental uncertainty. Therefore, we did not add any SVOC mass to these emissions. For gasoline sources, the data indicate an average bias of 40 %, which means that lower-volatility organics were only partially collected by the filter. This is consistent with the relatively low particle emissions of gasoline sources, which create lower concentration conditions inside of the dilution sampler, and therefore gas–particle partitioning shifted more to the vapor phase. We therefore applied a filter artifact correction factor of 1.4 to gasoline POA emissions, as shown in Table 3. We add these SVOC vapors to address the bias in emission measurements and to best estimate the potential local/regional SOA formation from mobile-source SVOCs.

## 3.3 IVOC emissions

An important difference from previous implementations of CMAQ to simulate the CalNex campaign (Baker et al., 2015; Murphy et al., 2017; Woody et al., 2016) is the new mobile IVOC emission data and the application of the new six-lumped-IVOC-species SOA parameterization. Mobile sources contribute more than 40 % of anthropogenic NMOG emissions in the South Coast Air Basin in the CalNex emission inventory (Baker et al., 2015). Given the consistency of the speciation and IVOC-to-NMOG ratio for sources using same type of fuel (Lu et al., 2018), we assign mobile-source emission profiles based on fuel type (gasoline, diesel or jet fuel). NMOG emissions from all on- and off-road gasoline sources are represented using the same average gasoline exhaust profile (SPECIATE profile no. 100VBS). NMOG emissions from all on-road and off-road diesel sources (including rail) are represented using the same average non-DPF (diesel particulate filter) diesel exhaust profile (SPECIATE profile no. 103VBS). Studies have noted there can be significant differences in IVOC emissions between DPF-equipped and non-DPF vehicles (Dunmore et al., 2015; Lu et al., 2018; Platt et al., 2017). However, the total NMOG emissions from diesel sources in southern California in 2010 were dominated ($> 99$ %) by non-DPF vehicles (due to a

combination of the fleet composition and the fact that non-DPF vehicles have much lower emission factors). Therefore, we use the IVOC emission profile for non-DPF vehicles for all diesel sources. Although only limited data are available for off-road diesel engine emissions (Qi et al., 2019), it suggests the emissions are similar to on-road diesel vehicles. NMOG emissions for all jet-fueled sources are represented using the same gas-turbine exhaust profile (SPECIATE profile no. 102VBS). The IVOC components of these profiles are summarized in Table 1, and complete profiles are given in SPECIATE 5.0 (US EPA, 2019). Total IVOC emissions are determined using the IVOC-to-NMOG ratios, which are more consistent across source types than IVOC-to-POA ratios (Lu et al., 2018).

For this work IVOC emissions are added to existing NMOG emissions. This was done to keep the VOC emissions across the different model runs constant in order to better isolate the contribution of IVOCs to SOA. In addition, OH oxidation of IVOCs is assumed to regenerate OH radicals and thus have minimal impact on the oxidant budget and the production of $O_3$. However, Lu et al. (2018) argued that existing NMOG inventories largely include IVOCs, just that they are misattributed to VOCs. Therefore, future work should proportionally reduce the VOC emissions to keep the overall NMOG emissions (VOC + IVOC) constant. This assumption minimally effects the OA model evaluation, because the base version of CMAQ predicts that traditional VOCs only contribute 7 % of measured OA in Pasadena during the CalNex campaign (Baker et al., 2015).

SOA is produced from IVOC oxidation using the parameterization described in Sect. 2. The SOA mass is determined by CMAQ based on the gas–particle partitioning of the SVOC products created from IVOC oxidation. CMAQ v5.3 calculates partitioning assuming thermodynamic equilibrium and that all organics form a single pseudo-ideal solution. The SVOC products also undergo multigenerational aging following the approach of Murphy et al. (2017) (see Sect. 3.4).

### 3.4   Multigenerational aging and gas–particle partitioning

The semivolatile POA emissions and semivolatile products formed from oxidation of SOA precursors undergo multigenerational aging as described in Murphy et al. (2017). Figure S5 shows the schematic diagram for modeling OH oxidation first-generation and multigenerational aging. Briefly, the approach simulates the reaction of L/S/IVOC vapors with hydroxyl radical and distributes the product mass to a second set of five vapor–particle pairs of species at moderate O : C values. The stoichiometric ratios used to distribute the product mass were derived to match the SOA enhancement predicted by a full 2D-VBS simulation of the functionalization and fragmentation of SVOCs during three days of atmospheric oxidation. This model, unlike that of Koo et

al. (2014), does transfer some of the aged products to higher-volatility bins and thus reduces SOA over multiple generations of OH reaction. The probability for fragmentation increases as a function of O : C in agreement with theory (Donahue et al., 2011). Although the competing effects of fragmentation and functionalization at long timescales are represented in this model, the simplified framework is likely limited when trying to capture the full complexity of multigenerational aging. For this work, no changes were made to the chemical properties (e.g., carbon number and O : C) or reaction stoichiometry of the multigenerational aging mechanism of Murphy et al. (2017). Because IVOC products likely have lower carbon numbers than products of primary SVOC oxidation, our approach may represent an upper bound on the potential for IVOC SOA aging to further enhance particle mass downwind of sources.

### 3.5   Simulation cases

To systematically explore the effects of adding IVOC emissions from mobile and nonmobile sectors, we performed four simulation cases, summarized in Table 4. All cases use the same emission inputs as described earlier with differences in IVOC emissions. In the base case (Case 1), mobile SOA is only formed through the oxidation of traditional VOC emissions and SVOCs from evaporated semivolatile POA.

Figure 3a compares the anthropogenic NMOG emissions in the Los Angeles Basin region for the four simulation cases (geographical boundaries are defined by simulation grid cells shown in Fig. S6). In the base case (Case 1), mobile sources contribute 43 % of anthropogenic NMOG emissions, of which gasoline sources contribute 35 %, diesel sources 8 % and aircraft less than 1 %. Nonmobile sources contribute the remainder of the anthropogenic NMOG emissions (57 %), of which volatile chemical product (VCP) usage contributes 39 %, followed by 17 % from other sources. The emission inventory contains minimal cooking and biomass-burning NMOG emissions during CalNex (1.5 %).

Cases 2 to 4 incrementally add mobile IVOC emissions to the model. Table 4 shows that Case 2 adds on average $27.6\,t\,d^{-1}$ mobile-source IVOC emissions, which is our best estimate of the mobile-source IVOC emission based on the compilation of measurement data and source profiles in Lu et al. (2018) as described in Sect. 3.3. The difference in SOA concentrations between Case 2 and Case 1 is the SOA contribution from mobile emitted IVOCs. In Case 3 and 4, we incrementally add IVOC emissions from nonmobile sources to the inventory to explore the contribution of nonmobile sources of IVOCs as discussed in Sect. 4.2.

## 4   CMAQ simulation results

To evaluate model performance, we compared predictions to measured data from the CalNex campaign in Pasadena,

**Table 4.** Total anthropogenic organic emissions (Ton day$^{-1}$) in the Los Angeles Basin region in four CMAQ simulation cases.

| Case | Name | Inventory POA | POA after scaling | Inventory NMOG | Mobile IVOCs | Nonmobile IVOCs |
|---|---|---|---|---|---|---|
| 1 | Base | 26.4 | 28.9 | 450.2 | 0 | 0 |
| 2 | Mobile IVOC | 26.4 | 28.9 | 450.2 | 27.6 | 0 |
| 3 | Low nonmobile IVOC | 26.4 | 28.9 | 450.2 | 27.6 | 30.7 |
| 4 | High nonmobile IVOC | 26.4 | 28.9 | 450.2 | 27.6 | 68.5 |

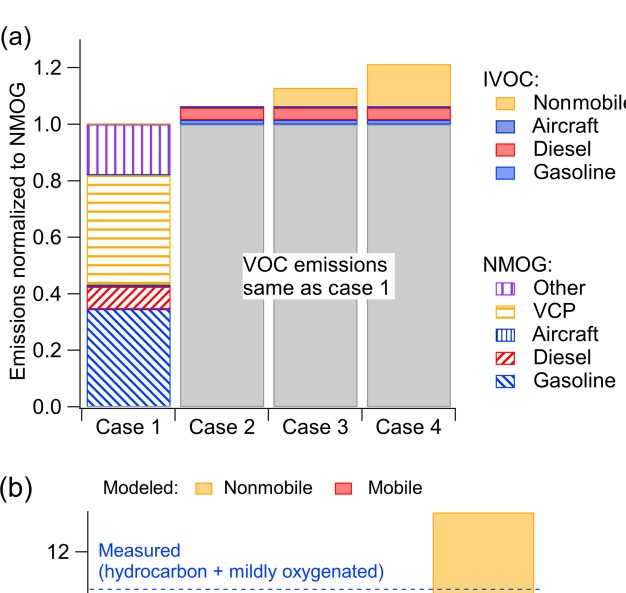

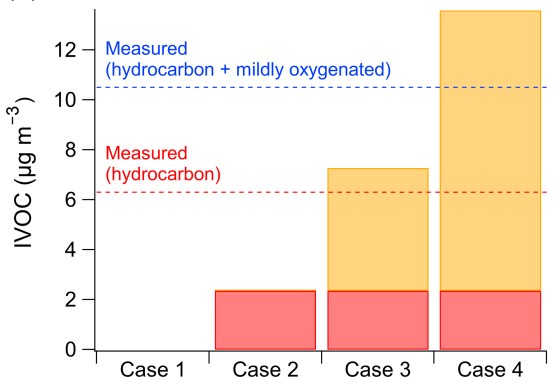

**Figure 3. (a)** Modeled NMOG and IVOC emissions by source for the four simulation cases. **(b)** Measured and modeled IVOC mass concentrations in Pasadena, CA, during CalNex for the four simulation cases. Measured data in **(b)** from Zhao et al. (2014). TS2

CA, as well as the organic carbon (OC) measured at Chemical Speciation Network (CSN) sites in California. The CalNex campaign characterized atmospheric composition at two sites in southern California, Pasadena and Bakersfield, from 15 May to 29 June 2010 (Ryerson et al., 2013). We focus on the Pasadena site, which is located 18 km northeast and generally downwind of downtown Los Angeles, because there were direct measurements of IVOCs (Zhao et al., 2014). We also evaluate model predictions at the Pasadena site for OA, BC, CO, select speciated VOCs and planetary boundary layer (PBL) height.

## 4.1 Base case and mobile IVOC case

### 4.1.1 IVOC mass concentrations

Figure 3b compares the model-predicted and measured campaign-average IVOC mass concentration at the Pasadena site. Zhao et al. (2014) reported data for two classes of IVOCs differentiated based on mass spectral signature: hydrocarbon IVOCs and mildly oxygenated IVOCs. Zhao et al. (2014) attributes hydrocarbon IVOCs to primary emissions; the mildly oxygenated IVOC could either be primary emissions or formed via atmospheric oxidation. The CalNex campaign-averaged measured hydrocarbon IVOCs at the Pasadena site were 6.3 µg m$^{-3}$; the measured mildly oxygenated IVOC concentration was 4.2 µg m$^{-3}$. The analytical techniques of Zhao et al. (2014) are not optimized for measuring oxygenated organics; therefore, their data provide a lower-bound estimate of the total and oxygenated IVOCs.

The base case (Case 1) predicts essentially no IVOC concentrations as they are not explicitly included in the base inventory or model (though could be implicitly included as misclassified VOC species). Case 2 (mobile IVOC case) predicts 2.4 µg m$^{-3}$
blackboxPlease give an explanation of why this needs to be changed. We have to ask the handling editor for approval. Thanks.**Might need approval by the editor.** of IVOCs at the Pasadena site, which corresponds to 38 % TS3 of measured hydrocarbon IVOCs. This indicates that mobile sources are an important source of IVOCs in the Los Angeles region but that more than half of the hydrocarbon IVOCs measured in Pasadena are likely emitted by nonmobile sources. In addition to hydrocarbon IVOCs, Zhao et al. (2014) measured 4.2 µg m$^{-3}$ of mildly oxygenated IVOCs, which are also not explained by mobile-source emissions.

While the comparison in Fig. 3b suggests that nonmobile sources may be important contributors to ambient IVOC concentrations, there are a number of potential uncertainties, including (1) uncertainty in mobile-source activity, (2) uncertainty in mobile-source NMOG emission factors, and (3) uncertainty in mobile-source IVOC-to-NMOG emission ratios. The first potential uncertainty is mobile-source activities. BC and CO are commonly used as indicators of gasoline and diesel source activity. The mobile-source CO emission inventory used here (EMFAC) agrees with another fuel-based CO

inventory (Kim et al., 2016), both of which reproduce the observed weekly patterns. This suggests the mobile-source CO emission inventory in the Los Angeles Basin during CalNex is correctly modeled. While the model performs well for CO (Fig. S2), it overestimates BC concentrations by a factor of 2. These comparisons suggest that gasoline activity (the major of source of CO) is modeled correctly, but there may be a potential overestimation of either diesel activity and/or the diesel BC emission factor (the major source of BC). If the diesel activity is overestimated, then diesel IVOC are likely overestimated, which only strengthens our conclusion that there are important nonmobile sources of IVOCs.

The second potential uncertainty is mobile-source NMOG emission factors. Comparisons in May et al. (2014) suggest that the EMFAC emission factors (which are used to create the mobile-source emission inventory for these simulations) are robust, except for LEV-II vehicles. During the 2010 CalNex period, EMFAC estimates LEV-II vehicles (considering model year after 2004) only emit 8.5 % of total gasoline NMOG emissions in California and therefore are not major contributors in mobile emissions. Therefore this uncertainty also does not appear to alter our conclusion that there are important nonmobile sources of IVOCs.

The final potential uncertainty is the IVOC-to-NMOG ratios. Zhao et al. (2016) and Lu et al. (2018) show that IVOC-to-NMOG ratios of cold-start UC (unified cycle) emissions from gasoline sources are consistent across a large number of vehicles spanning a range of emission certification standards. Although IVOC emissions from hot-running gasoline vehicle exhaust are enriched by as much as a factor of 4 compared to the cold-start UC cycle (Lu et al., 2018; Zhao et al., 2016), EMFAC2017 estimates that running exhaust only contributes 34 % of total gasoline summertime NMOG emissions in California in 2010. A simple weighted average of 66 % of emissions using the cold-start UC emission profile and 34 % of emissions using the hot-running emission profile increases the IVOC-to-NMOG fraction for gasoline vehicles by a factor of 2, from 4.5 % to 9.1 %. The IVOC-to-NMOG ratio for diesel sources is already high (55 %), and thus it cannot be increased as much as the gasoline emissions (less than a factor of 2). Therefore, the largest uncertainty in modeled mobile IVOCs is the gasoline source IVOC-to-NMOG ratio, which could be underestimated by as much as a factor of 2. This means that the overall uncertainty in modeled mobile IVOC emissions is less than a factor of 2. Increasing the gasoline IVOC emissions to better account for hot-running operations would explain a larger fraction of the measured hydrocarbon IVOC concentrations, but it seems unlikely that it would close the mass balance given that gasoline vehicles contribute less than half of the mobile IVOCs. Therefore, even acknowledging the existing uncertainty we still conclude that nonmobile sources are likely important contributors to ambient IVOC concentrations in Pasadena.

Jathar et al. (2017) also updated CMAQ with mobile-source IVOC emission estimates. They assumed that IVOCs contribute 25 % and 20 % of the NMOG emissions from gasoline and diesel sources, respectively. However, these ratios are not based on direct measurements but instead inferred from SOA closure studies for chamber experiments. The model of Jathar et al. (2017) predicted mobile sources contribute $3.9\,\mu g\,m^{-3}$ of IVOCs, which is about factor of 1.5 higher than the IVOC concentrations predicted here (and about 65 % of measured ambient hydrocarbon IVOC concentrations). The better closure is due to the very high IVOC-to-NMOG ratio assumed for gasoline vehicles, which is not supported by direct measurements (Drozd et al., 2019; Zhao et al., 2016).

### 4.1.2 Primary VOC/IVOC diurnal patterns

Figure 4 compares the measured and modeled campaign-average diurnal patterns of important anthropogenic VOCs (benzene, toluene, $m$-/$p$-/$o$-xylenes) and hydrocarbon IVOCs. Measured concentrations of benzene, toluene and hydrocarbon IVOCs are highest in the early afternoon (12:00–14:00, in Fig. 4a, b and d). This has been attributed to the transport of morning emissions from downtown Los Angeles to Pasadena (Borbon et al., 2013). Measured xylene concentrations show a slight decrease in daytime, which is attributed to their relatively high OH reaction rate and thus faster oxidation during the daytime (de Gouw et al., 2018).

Figure 4 indicates that the model reproduces the measured benzene diurnal pattern but not the toluene, xylene and hydrocarbon IVOC diurnal patterns. Figure 4b and c show that during nighttime the model overpredicts toluene and xylene concentrations by a factor of 2 and 1.4, respectively. Modeled hydrocarbon IVOC mass concentrations (Case 2) are underestimated throughout the day (Figs. 4d and 3b).

Figure 4 also shows modeled species concentrations peak around 06:00 and then steadily decrease from 06:00 to 14:00, in contrast to the early-afternoon peaks (12:00 to 14:00) in the measured data. A potential explanation for this difference is that the model is incorrectly simulating the PBL height. On average, the measured PBL height ranges from $\sim 200\,m$ at night to $\sim 900\,m$ at noon (Fig. S7), while modeled PBL height ranges from $\sim 60\,m$ at night up to 1500 m at noon. However, changing the predicted PBL height would degrade model performance for some species which are already predicted well (Figs. S3 and S4). Another possible explanation is that additional unknown sources of IVOCs have large NMOG emissions that peak at noon, for example some type of evaporative emissions. Additional research is needed to resolve the discrepancy between model and measured diurnal profiles shown in Fig. 4.

### 4.1.3 OA mass concentrations and diurnal patterns

Figure 5a plots the AMS-observed and CMAQ-modeled hourly-averaged $PM_1$-OA time series at the Pasadena site during CalNex. We consider the Pearson correlation coeffi-

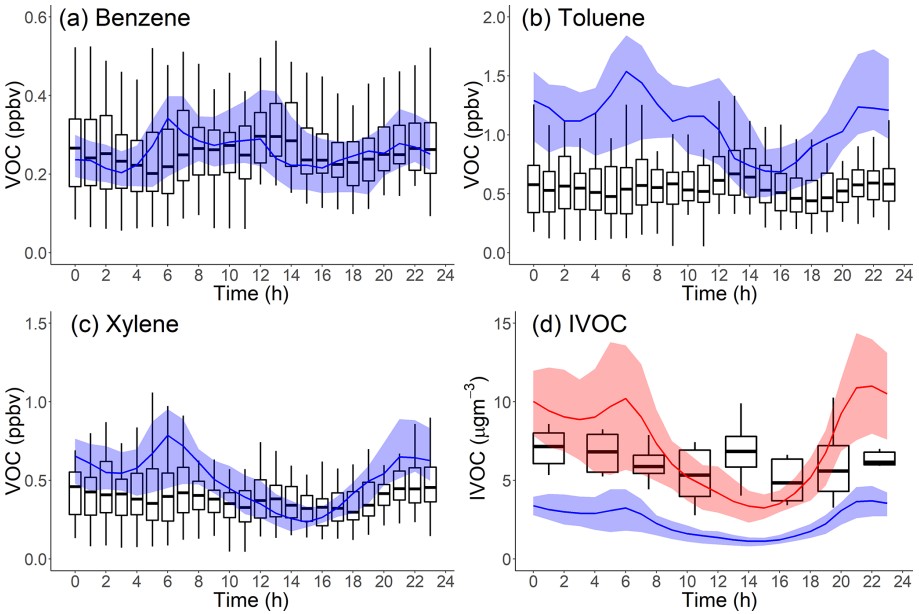

**Figure 4.** Comparison of measured (boxplot: solid box denotes 25th to 75th percentiles and whiskers denote 10th to 90th percentiles) and modeled (line: shaded area denotes 25th to 75th percentiles) diurnal patterns in Pasadena, CA, during CalNex for the following species: **(a)** benzene, $k_{OH} = 1.22 \times 10^{-12}$ cm$^3$ molec$^{-1}$ s$^{-1}$; **(b)** toluene, $k_{OH} = 5.63 \times 10^{-12}$ cm$^3$ molec$^{-1}$ s$^{-1}$; **(c)** xylene, $k_{OH} = 1.36 - 1.87 \times 10^{-11}$ cm$^3$ molec$^{-1}$ s$^{-1}$; and **(d)** hydrocarbon IVOCs (blue: Case 2, red: Case 3), $k_{OH} = 1.55 - 7.56 \times 10^{-11}$ cm$^3$ molec$^{-1}$ s$^{-1}$. Measured data from Borbon et al. (2013).

cient ($r$) and root-mean-square error (RMSE) as the evaluation metrics between measured and model OA time series. The definitions of $r$ and RMSE are shown in Eqs. (S1) and (S2) in the Supplement.

Our base model (Case 1) significantly underpredicts the OA concentration, often by more than a factor of 3, over the entire time period. Case 1 has a large RMSE = 5.3 µg m$^{-3}$, which is comparable to the average measured OA (6.9 µg m$^{-3}$), and moderate positive correlation ($r$ = 0.69). To understand the source of this discrepancy, Fig. 5b and c compare the modeled average diurnal patterns for SOA and POA to positive matrix factorization (PMF) factors derived from aerosol mass spectrometer data for OOA (SV-OOA plus LV-OOA) and POA (hydrocarbon organic aerosol (HOA) plus cooking organic aerosol (COA)) (Hayes et al., 2013). The observed OOA factor in Fig. 6b has a strong peak in the early afternoon, similar to the OH radical concentration (de Gouw et al., 2018) and photochemical age (Hayes et al., 2015).

Figure 5c shows that the model correctly predicts average POA concentrations (modeled: 1.73 µg m$^{-3}$ vs. measured: 2.01 µg m$^{-3}$). It also reasonably reproduces the observed POA diurnal pattern. This applies to all four cases and suggests that our inventory (Table 3) has a reasonable representation for the POA emissions, volatility distributions and correction for filter artifacts for gasoline sources. The mobile volatility profile predicts that a bit more than half of the semivolatile POA evaporates; therefore, if it treated POA

as nonvolatile then the model would have overpredicted the observed POA concentrations by about a factor of 2.

Figure 5b shows that Case 1 produces very little SOA, similar to previous CMAQ simulations (Baker et al., 2015; Woody et al., 2016). In this study, we emphasize the peak in the diurnal SOA concentration because this enhancement is reflective of the strength of prompt SOA formation in both the observations and the model. In Case 1, the predicted peak SOA concentration is 1.65 µg m$^{-3}$ at the Pasadena site, which is 5 times lower than the AMS-observed value (8.63 µg m$^{-3}$). Both modeled LV-OOA and SV-OOA are much lower than AMS-observed factors.

Figure 2 indicates that mobile-source IVOC emissions contribute significantly to SOA formation, especially to the daytime SOA formation due to their high SOA yield and OH reaction rates. In Case 2, the addition of mobile IVOC emissions increases the peak SOA concentration by 60 %, from 1.65 to 2.75 µg m$^{-3}$, and daytime SOA increases (peak SOA minus nighttime SOA) by 110 % from 0.82 to 1.73 µg m$^{-3}$. The increase in nighttime SOA from IVOC oxidation was about a factor of 4 smaller than the daytime increase. Adding mobile-source IVOC improves model performance, but Case 2 still only explained 32 % of AMS-observed daytime peak SOA.

Our comparison demonstrates that mobile-source IVOC emissions need to be explicitly included in models and inventories. However, they do not close the mass balance for hydrocarbon IVOCs or SOA in Pasadena. In the next sec-

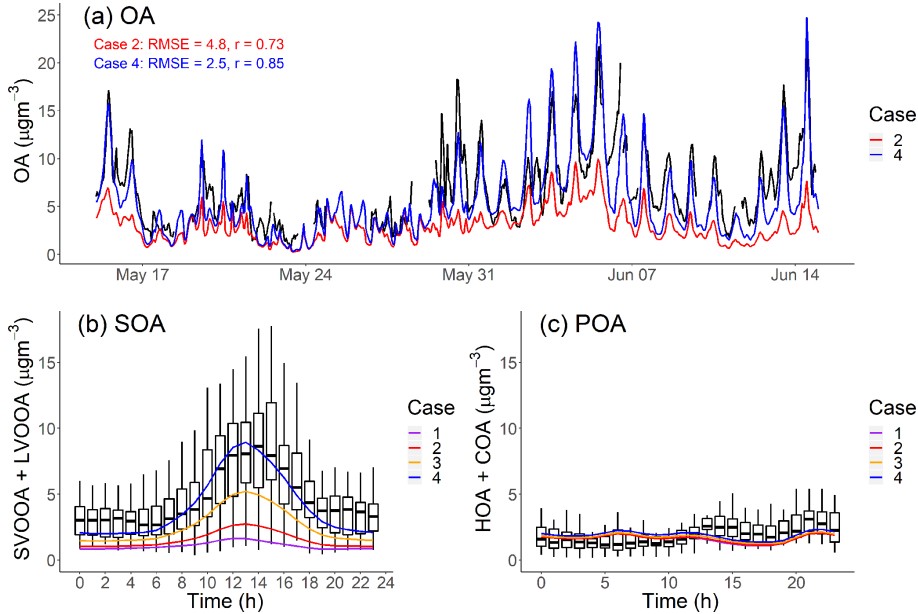

**Figure 5. (a)** PM$_1$-OA component hourly-averaged time series of measured data and model output in Pasadena, CA, during the CalNex campaign. **(b, c)** Diurnal pattern of measured and modeled SOA and POA mass concentration in Pasadena, CA, during CalNex. Measured data from Hayes et al. (2013).

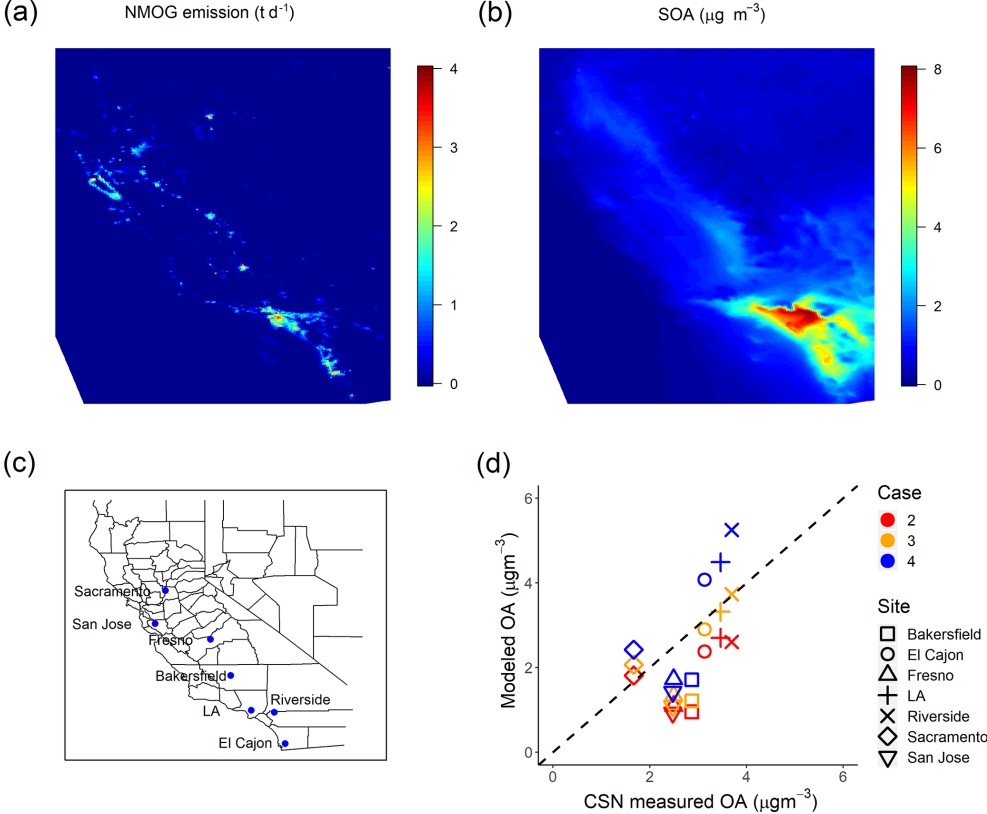

**Figure 6. (a)** Campaign-average NMOG emissions (t d$^{-1}$) in the emission inventory. **(b)** Modeled campaign-averaged SOA concentration in Case 4. **(c)** Location of CSN sites used for model evaluation. **(d)** Comparison of modeled OA to measured OA (OC·1.8) at CSN sites in California.

tion, we explore the potential contribution of IVOC emissions from nonmobile sources (McDonald et al., 2018).

## 4.2 Nonmobile IVOC emissions

### 4.2.1 IVOC mass concentrations and diurnal pattern

Motivated by recent research on volatile chemical products (VCPs) (Khare and Gentner, 2018; McDonald et al., 2018), we also investigated potential IVOC emission from nonmobile sources. For example, McDonald et al. (2018) estimated that 19.6 % of total gas-phase VCP emissions are IVOCs. Khare and Gentner (2018) reported that the IVOC content in 12 commercially available VCPs ranges from 0 % to 95 %. However, many of these IVOCs in VCPs are heavily oxygenated.

Cases 3 and 4 explore different levels of IVOC emission for nonmobile sources. The IVOC-to-NMOG ratios are not based on independent laboratory data but are set to close the gap between modeled and measured hydrocarbon IVOC concentration (Case 3) and SOA concentration (Case 4) in Pasadena, CA (Hayes et al., 2013; Zhao et al., 2014). Since there are limited data on nonmobile IVOC emissions, they are assumed to have the same properties as alkane-like IVOCs (IVOCP6-ALK to IVOCP3-ALK) with a uniform volatility distribution. Table 4 shows that Case 3 and 4 add an average of 30.7 and 68.5 t d$^{-1}$ of nonmobile IVOC emissions scaled from NMOG emissions as described in Sect. 3.4.

For the low nonmobile IVOC case (Case 3), we added IVOC emissions to the inventory equivalent to 12 % of nonmobile NMOG emissions. The scaling coefficient was determined to roughly match the campaign-average hydrocarbon IVOC mass concentrations measured in Pasadena, CA (Zhao et al., 2014). The only difference between Case 2 and 3 is the additional nonmobile hydrocarbon IVOC emissions.

For the high nonmobile IVOC case (Case 4), we added nonmobile IVOC emissions equivalent to 26.8 % of nonmobile NMOG emissions. This value was chosen to roughly close the mildly oxygenated IVOC and SOA mass balance. It is obviously a very high estimate but only somewhat higher than the 20 % estimates of total VCP emissions in McDonald et al. (2018). The only difference between cases 4 and 3 is the additional nonmobile IVOC emissions equivalent to 14.8 % of nonmobile NMOG emissions.

Figure 3b shows that in Case 3 the model predicts 4.9 μg m$^{-3}$ of nonmobile hydrocarbon IVOCs and 7.3 μg m$^{-3}$ TS4 of total hydrocarbon IVOCs, which is only somewhat higher than the measured value (6.3 μg m$^{-3}$). Case 4 predicts an additional 6.3 μg m$^{-3}$ of oxygenated IVOCs from nonmobile sources and 13.6 μg m$^{-3}$ TS5 of total IVOCs (hydrocarbon + oxygenated), which exceeds the measured total IVOC (10.5 μg m$^{-3}$) by 30 %. Given this overprediction and the fact that mildly oxygenated IVOCs can also be formed through secondary chemistry, these results suggest that the IVOC-to-NMOG ratio for nonmobile sources is be-

**Table 5.** Model OA performance metrics at all CSN sites (1.8· OC) for this study.

| Case | Fractional bias | Fractional error |
|---|---|---|
| 1 (baseline) | −0.59 | 0.67 |
| 2 (mobile IVOC) | −0.52 | 0.62 |
| 3 (low nonmobile IVOC) | −0.33 | 0.49 |
| 4 (high nonmobile IVOC) | −0.10 | 0.42 |

tween Case 3 (12 %) and Case 4 (26.8 %). In addition, recent research suggests that up to a factor of 3 scale-up may be needed for VCP NMOG emissions (McDonald et al., 2018), which would drive down the IVOC-to-NMOG ratios to 4 %–9 %.

### 4.2.2 OA time series and diurnal patterns

Adding nonmobile IVOC emissions increased the predicted afternoon peak SOA concentration to 5.0 and 8.6 μg m$^{-3}$ for Case 3 and 4, respectively. This highlights the potentially large contribution of nonmobile IVOC emissions to SOA formation. Figure 5a also shows that in Case 4 the modeled OA time series largely explains the observed SOA (RMSE = 2.5 μg m$^{-3}$, $r = 0.85$), including explaining the observed peak values in the middle of the day. Since increasing OA concentrations also shifts the gas–particle partitioning of SVOCs to the particle phase (Donahue et al., 2006), there are also minor shifts in POA partitioning from Case 1 to Case 4, but these changes are small and do not substantively alter the model–measurement POA comparison shown in Fig. 5c.

Adding nonmobile IVOC emissions also improves the model predictions of SOA contribution to OA in Pasadena. Hayes et al. (2013) apportioned 66 % of the OA to OOA (SV-OOA plus LV-OOA) in Pasadena during the CalNex campaign. Hersey et al. (2011) apportioned an even higher fraction of 77 % OA to OOA in Pasadena in 2009. As a comparison, if no IVOCs are included in the model, Case 1 only predicts that SOA only contributes 47 % of the total OA. With additional mobile and nonmobile IVOC emissions, our model predicts 67 % of OA as SOA in Case 3 and 74 % in Case 4.

Although Case 4 largely reproduces the measured OA, we do not think that missing IVOC emissions are the only contributor to the poor performance of the base model. The assumption of Case 4 that IVOCs contribute 26.8 % of nonmobile NMOG emissions is likely too high, and it overpredicts the total measured IVOC concentrations. Other important uncertainties include (1) the effect of vapor wall loss on SOA yield (Zhang et al., 2014), (2) PBL modeling, (3) multigenerational SOA aging and (4) SVOC emission uncertainties. First, SOA yields for VOCs and IVOCs need to be corrected (typically increased) for vapor wall losses (Akherati et al.,

2019). Second, CMAQ likely overpredicts the afternoon PBL height in Pasadena, as discussed in Sect. 4.1. Correcting this will likely increase SOA formation and concentrations, reducing the amount of IVOC emissions needed to reach SOA mass closure. Finally, the effects of multigenerational aging on secondary products of SOA precursor oxidation are uncertain. We have represented this phenomenon with model parameters designed for aging of SVOC emissions (Murphy et al., 2017), but the ratio of functionalization versus fragmentation could be different for products of IVOC oxidation due to differences in carbon number and functionality. Figure 5c shows that simulated POA reproduces the measured concentrations, so we believe that the uncertainty in SVOC emissions is relatively small. We also acknowledge the model uncertainty in the oxidation and aging of SVOCs, and this can lead to the substantial changes in OA prediction.

Despite all of these potential uncertainties, the exploratory simulations (Case 3 and 4) indicate nonmobile IVOC emissions are likely an important source of SOA precursors, but their contributions should be between Case 3 and 4 (12 % and 26.8 % of nonmobile NMOG emissions). The lower value will close the hydrocarbon IVOC but not the SOA mass balance. Correcting the likely underestimate of VCP emissions (McDonald et al., 2018) in current inventories will drive down the needed nonmobile IVOC emissions from 4 % to 9 % of NMOG emissions.

## 4.3 Regional SOA formation

IVOCs also contribute to regional SOA formation. This is shown in Fig. 6a and b, which present maps of campaign-average NMOG emissions and modeled SOA concentrations. Primary NMOG emissions are concentrated in densely populated urban areas such as Los Angeles, but due to the transport of SOA precursors, especially IVOCs, Fig. 6b shows that SOA concentrations are spread over a much larger spatial domain than the emissions. This is expected given the SOA production requires time for atmospheric oxidation.

To evaluate the spatial performance of the model, we compared model predictions of regional OA to CSN data at seven sites in California shown in Fig. 6c. Three of the sites are in southern California (Los Angeles, Riverside and El Cajon), while the others are in central or northern California. Figure 6d shows the comparison between modeled OA and CSN data (OC·1.8 to account for non-carbonaceous components of the organic aerosol collected on the filters) for all seven sites from Case 1 to Case 4. Table 5 summarizes the evaluation metrics for all cases in site-aggregated comparisons.

Case 1 grossly underestimated the OA at all sites except for Sacramento, with a fractional bias (FB, definition in the Supplement) of $-0.59$ and fractional error (FE, definition in the Supplement) of 0.67, of which much of the measured OA is SOA (Docherty et al., 2008; Hayes et al., 2013). Case 2 and Case 3 reduce the fractional bias to $-0.52$ and $-0.33$, respectively, and the fractional error to 0.62 to 0.49. Of the

four cases considered here, Fig. 6d shows that Case 3 predicted the OA concentrations at three of the southern California CSN sites but underpredicts at other sites such as Fresno, San Jose and Bakersfield. Case 4 overpredicts the OA concentrations at the southern California CSN sites (coincident with the highest average SOA concentrations) but still underpredicts in Bakersfield, San Jose and Fresno. However, this case has the best overall metrics (FB = $-0.10$ and FE = 0.42).

Figure 6b shows that the amount of SOA formed from additional IVOC emissions is much less in northern and central California compared to southern California. This could be due to the different meteorological conditions, or source variations, and/or inaccuracies in the multigenerational aging model. More research is needed to better understand the competition between functionalization and fragmentation of organic gases at long atmospheric timescales. Case 3 and Case 4 were estimated to roughly explain the measured hydrocarbon IVOC and SOA concentration in Pasadena, but measured data of source-specific IVOC-to-NMOG fractions are needed to correctly model the nonmobile emissions.

## 5 Conclusions

This paper presents new mobile-source emission profiles that explicitly account for IVOC emissions and a new SOA parameterization for mobile-source IVOCs designed for implementation in chemical transport models. We implemented these new profiles and parameterizations to investigate the contribution of mobile sources and IVOC emissions to SOA formation in California during the CalNex campaign. We have focused on mobile-source emissions because of the availability of data, but the same basic approach can applied to other sectors of organic combustion in the future, such as wildfires, agricultural fires and meat cooking, as additional data become available. The main findings are as follows.

We developed a new parameterization to model SOA formation from mobile-source IVOC emissions designed for implementation into CTMs. Explaining the SOA formation from both gasoline and diesel vehicles requires accounting for both the volatility and the chemical composition of the IVOC emissions. Our parameterization has six lumped IVOC species: two aromatic and four aliphatic.

We developed new source profiles for IVOC emissions from mobile sources that are available in SPECIATE 5.0 to facilitate their use in emission inventory preparation and future CTM simulations. Applying these profiles to the existing EPA inventories predicts that mobile sources contribute 2.4 µg m$^{-3}$ [TS6] of IVOCs at the Pasadena site during CalNex, which is 38 % [TS7] of measured concentrations of hydrocarbon IVOCs.

Mobile-source IVOC emissions are predicted to contribute $\sim 1$ µg m$^{-3}$ daily peak SOA concentration, a 67 % increase compared to the base case without IVOC emissions. There-

fore, mobile-source IVOC emissions need to be included in CTM simulations. However, mobile-source emissions alone do not explain the measured IVOC or SOA concentrations. The growing importance of nonmobile sources underscores the effectiveness of the decades-long regulatory effort to reduce mobile-source emissions. Results from exploratory model runs suggest that between 12 % and 26.8 % (or 30.7 to 68.5 t d$^{-1}$ in the Los Angeles–Pasadena region) of nonmobile NMOG emissions are likely IVOCs.

Future research needs the following.

– VCPs are likely a major source of IVOCs and future research is needed to constrain their emissions using ambient observations, bottom-up emission inventory methods and computational models (McDonald et al., 2018; Qin et al., 2020). Measurements of both the volatility distribution and chemical composition of VCP emissions are needed. Modeling the SOA formation from these new IVOCs will likely require extension of existing chemical mechanisms to better represent more oxygenated IVOCs.

– More measurements of ambient IVOC concentrations across a range of field sites are needed to better evaluate model performance. Given the lack of data, regional evaluations of ambient IVOC and OA predictions still have large uncertainty.

– Improved understanding is needed on the effects of multigenerational aging on SOA formed from IVOC emissions (and other precursors). The impacts of polluted plumes on downwind receptors depend on the nature of aging processes and whether they result in the addition or reduction of particulate mass (e.g., fragmentation processes may enhance volatilization of OA downwind of sources).

*Data availability.* Source emission profiles used in this work are available in SPECIATE 5.0. CalNex measurement data are available through the Earth System Research Laboratory data portal (https://www.esrl.noaa.gov/csd/groups/csd7/measurements/2010calnex/, CalNex measurement data, 2012 TS8). Data under the figures are available at the ScienceHub database (https://doi.org/10.23719/1505451 TS9). Model output data files are available upon request to the contact author. CE4

*Supplement.* The supplement related to this article is available online at: https://doi.org/10.5194/acp-20-1-2020-supplement.

*Author contributions.* QL, BNM and ALR designed the research. QL, BNM, PJA, YZ and ALR designed the SOA formation parameterizations from IVOCs and carried them out. QL, BNM, MQ, PJA, HOTP, CE, CA and ALR designed the CMAQ model configuration and simulations and carried them out. QL wrote the paper with input from all coauthors.

*Competing interests.* The authors declare that they have no conflict of interest.

*Disclaimer.* Although portions of this work were contributed by research staff in the Environmental Protection Agency and this work has been reviewed and approved for publication, it does not reflect official policy of the EPA. The views expressed in this document are solely those of authors and do not necessarily reflect those of the Agency. EPA does not endorse any products or commercial services mentioned in this publication. CE5

*Acknowledgements.* This publication was developed as part of the Center for Air, Climate, and Energy Solutions (CACES). The authors thank CACES team members and Neil M. Donahue for helpful discussions.

*Financial support.* This research is supported under assistance agreement no. RD83587301 awarded by the U.S. Environmental Protection Agency. QY was also partially supported by the Oak Ridge Institute for Science and Education (ORISE) Research Participation Program for the U.S. Environmental Protection Agency (EPA).

*Review statement.* This paper was edited by Manabu Shiraiwa and reviewed by two anonymous referees.

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

**Remarks from the language copy-editor**

CE1    This figure has been reprocessed following your clarification. Please verify the figure contents carefully.

CE2    Please confirm change.

CE3    It's our standard to abbreviate SI-accepted units of measure up to day (i.e., s, min, h, and d).

CE4    Please confirm new section.

CE5    Please verify the changes to the disclaimer, financial support section, and acknowledgements. I moved the last part of the financial support section you proposed to this section as the financial support section is restricted to funding information.

**Remarks from the typesetter**

TS1    Thank you for the new key figure. Please confirm that the copyright of this figure is still "© Author(s). Distributed under the Creative Commons Attribution 4.0 License. " or provide the correct one.

TS2    Please give an explanation of why the value in the figure needs to be changed. We have to ask the handling editor for approval. Thanks.

TS3    Please give an explanation of why this needs to be changed. We have to ask the handling editor for approval. Thanks.

TS4    Please give an explanation of why this needs to be changed. We have to ask the handling editor for approval. Thanks.

TS5    Please give an explanation of why this needs to be changed. We have to ask the handling editor for approval. Thanks.

TS6    Please give an explanation of why this needs to be changed. We have to ask the handling editor for approval. Thanks.

TS7    Please give an explanation of why this needs to be changed. We have to ask the handling editor for approval. Thanks.

TS8    Please provide this citation.

TS9    Please check DOI and provide a citation.