# Peer review of "Simulation of organic aerosol formation during the CalNex study: updated mobile emissions and secondary organic aerosol parameterization for intermediate volatility organic compounds"

_Atmospheric Chemistry and Physics, 2019_

## Referee Comment (RC1) · Anonymous Referee #1 · 18 Dec 2019

This paper discusses the hot-topic of IVOC emissions, making use of detailed emission inventory data and a new approach to SOA modelling. Overall the paper is well written, and the methods and asssociated data and assumptions well described.

Some issues which should be addressed are:

* The discussison of SOA yields and reactivity seems to be entirely focused on OH reactions. What about O3 and NO3 reactions?

[Figure]

* On page 4 it says that IVOC emissions are modelled based upon fuel type usage, but what about the emission control technology? I note that Lu et al (2018) commented that emissions show a consistency across sources for the same fuel type, but studies in other areas would seem to offer differing conclusions (cf. Platt et al. 2017 vs Dunmore et al, 2015 with respect to diesels.)

* The whole discussion is about IVOC. Cannot SVOC uncertainties also explain some of the results?

* The model evaluation is focused on just one site, Pasadena, except for some OA comparisons for six sites in Fig. 6. Although there may be good reasons to focus on Pasadena, the model results for other sites for pollutants such as CO or NOx should be briefly presented (maybe in SI). I don't think one can make very strong conclusions about the 'accuracy' or reasonableness of a SOA model based upon one's ability to reproduce observations at one site only, or of only SOA at six sites.

* The discussion of uncertainties from p10-12 was a bit confusing. Section 4, CMAQ simulation results, starts with predictions of IVOC at Pasadena. I would have preferred to see some model validation results first, before diving into the IVOC predictions. At this stage, we have no idea if the model works well for even the simple pollutants such as CO. Later (p12) we read about some NMOG comparisons, but those results are quite mixed (good for benzene, bad for toluene), so one can't be sure how accurate the NMOG inventory (or modelling) is. Page 11 manages to conclude that the uncertainty in IVOC is within a factor of 2 before these NMOG data are presented. Why, when some of the problems could reasonably be attributed to incorrect NMOG emissions? Also, p12 starts to attribute some problems to PBL height, but then the good agreement found for CO has to be explained.

Some small issues: —————————————

p2, L36, Abstract. With respect to SOA modelling, I am wary of phrases such as 'the most accurate photochemical model prediction'. It is always difficult to know if the

improved performance is really due to a more accurate modelling system, or getting things "right for the wrong reasons". While I think the authors approach is generally as sound as one can do with SOA modelling, I would anyway rephrase to say something like "model predictions which are the most consistent with observations to date", or something similar.

p4, L102. What is meant by the quoted word enrichment? That word doesn't appear in the first two papers cited in that sentence (I didn't check the others).

p7, L194. What does 'remade' mean here?

p7, L201-202. It is stated that wind-blow dust should not impact comparison with data collected by AMS, but dust impacts nitrate formation (e.g. Hodzic et al., 2006) and indeed many gas-particle processes.

p9, L255-259. I think another important reason to not reduce VOC emissions is that this would impact the O3 chemistry in an unrealistic fashion. At least, unless you allow your IVOC to undergo gas-phase reactions, which I don't think is the case.

p11, L308. Language - emission inventories cannot be "correctly modelled".

p12. I don't think equations 2 and 3 are really needed, or should be moved to SI, since these are standard statistical metrics. If included (e.g. in Supplementary), then you have to explain 'Cov', 'Var', and N. You should in case explain though if you are taking statistics over hourly values. (The equations for FB and FE on page 15 could also be moved to SI.

p12, L363. OOA is not really "measured" by AMS; it is estimated through the somewhat subjective procedures of PMF.

p13, L389. Why 'However,'?

p14. With regard to these evaluations, I also wonder why biomass-burning, or COA are not addressed in more detail?

p27, Table 2. This table gives the MW of the compounds. I am just curious - are the authors using these in their application of Raoult's law, or do they use the Donahue et al 2006 (Supplemental section) idea that one can use mass fraction rather than mole-fraction in the simulations?

p27, Table 2. Please be explicit that yields are mass-based.

p29, Table 4. The term traditional is starting to lose its meaning. Do you mean inventory or officially-reported POA?

p31, Figure 1. Give units.

Figs 2, 3, 6 and S1 had some formatting issues on my printer. Use a differnt graphics format if possible (.eps, .png?)

References

Dunmore et al., Diesel-related hydrocarbons can dominate gas phase reactive carbon in megacities, ACP, www.atmos-chem-phys.net/15/9983/2015/

Hodzic, A. et al, A model evaluation of coarse-mode nitrate heterogeneous formation on dust particles Atmos. Environ., 2006, 40, 4158-4171

---

## Referee Comment (RC2) · Anonymous Referee #2 · 14 Jan 2020

Summary: This study summarizes the development of a new SOA model and emissions inventory to explore the contribution of IVOCs to SOA formation in California. New IVOC source profiles are developed for gasoline engines, diesel engines, food cooking, wood smoke, and all other sources. A new VBS model is parameterized to predict SOA formation under typical conditions. Calculations are carried out using CMAQ for the region surrounding Los Angeles and for the entire state of California during the months of May-June 2010. The major conclusions of the study are that

(i) gasoline sources emit 4 times more NMOG than diesel sources, (ii) diesel emits 3x more IVOC than gasoline, (iii) mobile sources contribute ∼1 μg m-3 of SOA at Pasadena (incl IVOC contributions), (iv) the additional SOA from mobile source IVOCs does not close the gap with measured SOA concentrations, (v) missing IVOC sources cannot realistically be explained by mobile sources, and (vi) missing IVOC sources could plausibly be related to VCPs.

Comments:

1. Line 129: The constraint to limit the SOA model to fewer than 79 additional species seems artificial. While it is true that regulatory applications of CMAQ may not want to implement calculations with large numbers of species, exploratory scientific applications of CMAQ could easily add this number of species to the SOA model. Several examples have been published in the literature (see for example papers by Ying et al. using CMAQ).

2. Line 157, eq 1: The SOA mode formulation used in the current study does not allow for fragmentation. It may not be possible to fit the coefficients in the model to adequately explain SOA formation under the full range of atmospheric aging at different VOC/NOx ratios with this limitation. Some discussion of this limitation should be included in this section that introduces the simplified SOA model.

3. Line 194: change "and" to "by"?

4. Line 213: using diesel POA as a surrogate for all other combustion sources besides mobile, cooking, and biomass seems like a bit of a simplification. It may not matter much for the overall SOA analysis, but is this really appropriate for sources like aircraft? Or structure fires? Or natural gas combustion? Using diesel POA volatility for these sources could significantly bias the results around some localized sources.

5. Line 228: The authors go to great trouble to estimate the amount of additional POA that was not measured during emissions testing due to low concentrations in sampling

equipment, and then describe this material as semi-volatile using POA volatility distributions (line 208). It isn't clear that this gives a different (better) answer than just leaving the original POA emissions at their nominal values and describing this material as essentially non-volatile. It would be instructive to other readers if the authors could quantify these two treatments of POA (or refer to previous publications where this has been done and summarize the results).

6. Line 250: off-road gasoline does not account for a majority of emissions, but shouldn't off-road gasoline engines (non-catalyst) have their own unique profiles?

7: Line 251: off-road diesel is a major source of emissions. It definitely seems like a stretch to use the on-road diesel profile to describe off-road diesel emissions. The uncertainty introduced by this issue should be analyzed in the paper.

8. Line 262: Fragmentation and functionalization are accounted for, but in a biased manner. All of the SVOC and IVOC emitted in the current study will eventually make SOA due to the absence of fragmentation in the mechanism. The formation rates are tuned to account for the net effects over some pre-defined range of aging, but this simplistic model cannot capture the behavior correctly over the full lifetime. It is beyond the reasonable scope to change the simple model in the current paper, but the authors should properly describe it's limitations.

9. Line 312: EMFAC emissions factors are cited as a source of uncertainty, but line 199 states that mobile on-road and non-road emissions are calculated by MOVES 2014a. Which is it?

10. Line 325: The paper should also acknowledge that unknown chemical reactions leading to the formation and reaction of IVOCs could play a role in model error.

11. Figure 6 lower right panel: each of these sites has a different representative atmospheric aging time. The fact that some over-predict and some under-predict as the emissions are scaled up and down may reflect the fact that the functionalization / fragmentation processes have been combined into a single lumped parameter that works at one time scale but not at others. This possibility should be discussed in the results and model formulation sections.

---

## Author Comment (AC1) · 28 Feb 2020

We thank both reviewers for their careful reading and valuable comments. We have updated the manuscript based on their comments and provided a detailed response below. Reviewer comments are in regular black, our responses are in blue, and the additions/updated text from the manuscript are in red.

**Reviewer 1**

**General comments:**

\* The discussion of SOA yields and reactivity seems to be entirely focused on OH reactions. What about O3 and NO3 reactions?

Author's response: For oxidation of anthropogenic VOCs, we focused on OH oxidation for consistency with other studies using chemical transport models of varying scales (i.e. regional to global). In this study, we only account for IVOC-OH reactions because mobile-source IVOCs are mostly alkanes or aromatics, which will react slower with $O_3$. In addition, single ring aromatics (benzene, toluene, xylene) in existing chemical mechanisms only react with OH. Empirical representations of anthropogenic SOA that have successfully reproduced ambient SOA concentrations (e.g. SIMPLE by Hodzic and Jimenez 2011 or potential combustion SOA (Murphy et al. 2017)) focus exclusively on OH. Research shows that OH oxidation of ambient forest air led to approximately 4 times more SOA formation than either $O_3$ or $NO_3$ oxidation in oxidation flow reactor experiments (Palm et al., 2017). We recognize that for some VOCs, $NO_3$ oxidation is important in night-time SOA formation (Fry et al., 2014; Hoyle et al., 2011), and these will be important to consider in the future, but experimental studies on SOA formation from anthropogenic IVOC reactions with $NO_3$ radical are limited at this time.

Changes in the manuscript: p6, L151: add 'In this study, we only account for IVOC-OH reactions because mobile-source IVOCs are mostly alkanes or aromatics, which will react slower with $O_3$. $NO_3$ oxidation can be important in night-time SOA formation (Fry et al., 2014; Hoyle et al., 2011), and these will be important to consider in the future, but experimental studies on SOA formation from anthropogenic IVOC reactions with $NO_3$ radical are limited at this time.'

\* On page 4 it says that IVOC emissions are modelled based upon fuel type usage, but what about the emission control technology? I note that Lu et al (2018) commented that emissions show a consistency across sources for the same fuel type, but studies in other areas would seem to offer differing conclusions (cf. Platt et al. 2017 vs Dunmore et al, 2015 with respect to diesels.)

Author's response: For gasoline vehicles, we showed that the IVOC emissions are consistent across a wide range of model years in Lu et al. (2018). The SOA chamber study by Zhao et al. (2018) measured similar effective SOA yields across different engine technologies and certification standards, which also imply similar IVOC chemical composition. Saliba et al. (2017) showed that there is no statistical difference of regulated gas-phase pollutant emissions between PFIs and GDIs. Drozd et al. (2019) also showed that the composition of IVOC emissions was consistent across the test fleet from Tier-0 to SULEVs.

For diesel vehicles, we noticed the significant differences in IVOC emissions between DPF-equipped and non-DPF vehicles, similar to the findings in Platt et al. (2017) vs Dunmore et al. (2015). We included two separate emission profiles in Lu et al. (2018). But because the DPF vehicles emission factors are nearly two orders of magnitude lower than non-DPF vehicles, the total emission was still dominated (> 99%) by non-DPF vehicles in 2010, we use the IVOC emission profile of non-DPF vehicle as the surrogate.

Changes in the manuscript: p10, L296: add 'Studies have noted there can be significant differences in IVOC emissions between DPF-equipped and non-DPF vehicles (Dunmore et al., 2015; Lu et al., 2018; Platt et al., 2017). However, the total NMOG emission from diesel sources in southern California in 2010 were dominated (> 99%) by non-DPF vehicles (due to a combination of the fleet composition and the fact that non-DPF vehicles have much lower emission factors). Therefore, we use the IVOC emission profile for non-DPF vehicle for all diesel sources.'

* The whole discussion is about IVOC. Cannot SVOC uncertainties also explain some of the results?

Author's response: Yes, SVOC uncertainties definitely can explain some of the results. In this version of CMAQ, SVOCs are co-emitted and scaled from POA emissions, because they are the more substantial part of POA than NMOG emissions. Figure 5(b) shows that simulated POA agrees with the measurements, so we believe that the uncertainty in SVOC emissions are relatively small.

There's still uncertainty in the oxidation and aging of SVOCs, and this can lead to the substantial changes in OA prediction. We will include this uncertainty in the discussion.

Changes in the manuscript: p15, L438: add '(4) SVOC emission uncertainties'

p15, L444: add 'Figure 5(c) shows that simulated POA reproduces the measured concentrations, so we believe that the uncertainty in SVOC emissions are relatively small. We also acknowledge the model uncertainty in the oxidation and aging of SVOCs, and this can lead to the substantial changes in OA prediction.'

* The model evaluation is focused on just one site, Pasadena, except for some OA comparisons for six sites in Fig. 6. Although there may be good reasons to focus on Pasadena, the model results for other sites for pollutants such as CO or NOx should be briefly presented (maybe in SI). I don't think one can make very strong conclusions about the 'accuracy' or reasonableness of a SOA model based upon one's ability to reproduce observations at one site only, or of only SOA at six sites.

Author's response:

We have evaluated the $O_3$ and $NO_x$ predictions in Pasadena, and Figure S5 shows that the model well predicts the $O_3$ and NO diurnal patterns, and the $NO_2$ diurnal patterns in daytime. We also evaluated CO, $O_3$ and NO predictions in Bakersfield, Sacramento and Cool. Figure S6 shows that for Sacramento and Cool, CO, $O_3$ and NO predictions agree with the measurement data within ± 25% range, only except $O_3$ and NO prediction in Bakersfield. Therefore, for all four sites in California, we have shown strong correlation in gas-phase species (CO, $O_3$ and $NO_x$) between modelled and measurement values. Future work will evaluate the reasonableness of this SOA model at sites throughout the U.S. along with other model species. The current study's aim is to target a field data from a campaign for which there exists a strong record of analysis on sources and properties of SOA and IVOCs.

Changes in the manuscript:

p8, L232: add 'Previous studies (Baker et al., 2015; Woody et al., 2016) have extensively evaluated different versions of CMAQ using CalNex data. These evaluations show good to excellent performance for many pollutants, with a notable exception of organic aerosols and SOA – the focus of this paper. We evaluated our model predictions with measurements of gas-phase pollutants such as CO, $O_3$ and $NO_x$, as they are typical indicators for model performance. Consistent with the previous applications of CMAQ to

CalNex (Baker et al., 2015; Murphy et al., 2017; Woody et al., 2016), Figure S2(a) shows very good agreement between modelled and measured CO diurnal patterns in Pasadena, and the normalized mean bias (NMB) is 4.2%. Figure S3 compares the $O_3$, NO and $NO_2$ diurnal patterns with measurements in Pasadena, where the NMB is 10.7%, -6.7% and 5.4%, respectively. Figure S4 also compares the CO, $O_3$ and NO diurnal patterns for three other sites: Bakersfield, Sacramento and Cool. The model NMB is within ±25% for all comparisons, except for $O_3$ and NO in Bakersfield. Thus, we can conclude that the CMAQ model perform reasonably well at all four sites for traditional gas-phase pollutants.'

In the SI: add Figure S3 and S4.

[Figure]

Figure S3: Comparison of measured (boxplot, solid box denotes 25[th] to 75[th] percentiles and whiskers denote 10[th] to 90[th] percentiles) and modelled (line, from Case 1 to Case 4) diurnal patterns in Pasadena, CA during CalNex for species: (a) Ozone  (b) NO and (c) $NO_2$

[Figure]

Figure S4: Comparison of measured (boxplot, solid box denotes 25[th] to 75[th] percentiles and whiskers denote 10[th] to 90[th] percentiles) and modelled (line, shaded area denotes 25[th] to 75[th] percentiles) diurnal patterns during CalNex and CARES for species: CO, $O_3$ and NO in (a-c) Bakersfield, (d-f) Sacramento and (g-i) Cool

\* The discussion of uncertainties from p10-12 was a bit confusing. Section 4, CMAQ simulation results, starts with predictions of IVOC at Pasadena. I would have preferred to see some model validation results first, before diving into the IVOC predictions. At this stage, we have no idea if the model works well for even the simple pollutants such as CO. Later (p12) we read about some NMOG comparisons, but those results are quite mixed (good for benzene, bad for toluene), so one can't be sure how accurate the NMOG inventory (or modelling) is. Page 11 manages to conclude that the uncertainty in IVOC is within a factor of 2 before these NMOG data are presented. Why, when some of the problems could reasonably be attributed to incorrect NMOG emissions? Also, p12 starts to attribute some problems to PBL height, but then the good agreement found for CO has to be explained.

Author's response:

We now add the paragraph discussing model evaluation with measurement data of gas-phase pollutants such as CO, $O_3$ and $NO_x$, for all four sites. Then we present IVOC comparisons first because of the extensive body of literature already present comparing models CMAQ and other similar models to the measurements from the CalNex campaign (e.g. Baker et al. (2015), Woody et al. (2016), Murphy et al. (2017), Jathar et al. (2017)). The subsequent discussions of performance for other NMOG species, performance for CO, and behavior of the modeled PBL height are provided in order to contextualize the behavior of the IVOCs and indicate some potential areas for future improvement. Errors in mobile IVOC predictions are certainly attributable to varying degrees to errors in NMOG emission factors, mobile source activity, speciation, and meteorology.

As stated in p13, we are able to conclude that IVOC uncertainties are within a factor of 2 despite potential NMOG emission factor issues because independent literature (e.g. May et al., 2014) have confirmed that the emission factors from EMFAC used to inform the MOVES 2014 model are robust. At this point, there do appear to be performance inconsistencies among likely mobile-source primary pollutants like CO, benzene, toluene, xylenes, and BC. However, these inconsistencies vary from unbiased to overprediction by approximately a factor of 2. This would indicate that if potential biases in NMOG emission factors or mobile-source activity are resolved, it would tend to shift concentrations downward, and further supporting our conclusion that mobile-IVOCs are significant and important but not sufficient to account for all ambient IVOCs measured in Pasadena.

While emission inventory and meteorological issues will certainly impact IVOC performance for models like CMAQ, our primary motivation is to present this independently, rigorously verified dataset of IVOC speciation from NMOG and corresponding SOA yields and demonstrate how their impact on SOA performance in an off-the-shelf regulatory model with emissions prepared from standard methods. By keeping these other elements fixed, we provide a realistic picture of how modeled IVOCs will behave currently. As future improvements to emissions and meteorology occur, we will continue to evaluate and improve the uncertain SOA parameters.

Changes in the manuscript:

p8, L232: add the paragraph 'Previous studies … gas-phase pollutants.' as discussed above.

SI: add Figure S3 and S4 as discussed above.

**Some small issues:**

p2, L36, Abstract. With respect to SOA modelling, I am wary of phrases such as 'the most accurate photochemical model prediction'. It is always difficult to know if the improved performance is really due to a more accurate modelling system, or getting things "right for the wrong reasons". While I think the authors' approach is generally as sound as one can do with SOA modelling, I would anyway rephrase to say something like "model predictions which are the most consistent with observations to date", or something similar.

Author's response: We admit that CTMs are very complex models, and sometimes hard to claim accuracy before we know everything well. We will change accordingly.

Changes in the manuscript: change to 'By incorporating both comprehensive mobile-source emissions profiles for SVOCs and IVOCs and experimentally constrained SOA yields, this CMAQ configuration best represents the contribution of mobile sources to urban and regional ambient OA.'

p4, L102. What is meant by the quoted word enrichment? That word doesn't appear in the first two papers cited in that sentence (I didn't check the others).

Author's response: Enrichment means 'the relative fraction of a group of compounds increased after certain process'. We removed the phrase to avoid confusion.

p7, L194. What does 'remade' mean here?

Author's response: We remade the meteorology and emission inventory files for finer grid cells used in this work and Qin et al. (2019).

Changes in the manuscript: change 'remade' to 'identical to'

p7, L201-202. It is stated that wind-blow dust should not impact comparison with data collected by AMS, but dust impacts nitrate formation (e.g. Hodzic et al., 2006) and indeed many gas-particle processes.

Author's response:

While we agree with the author of the many interesting impacts of dust on fine and coarse PM components, previous studies found no evidence of excessive dust concentrations during the CalNex campaign (Cazorla et al., 2013) and we are focusing our analysis primarily on SOA generated promptly from sources within the L.A. basin. Future simulations with these mechanistic SOA pathways will include dust emissions and be able to better address these potentially important interactions.

Changes in the manuscript: p8, L225: add 'Moreover, previous studies (Cazorla et al., 2013) found little evidence of dust impacts during CalNex using both in-situ aircraft measurements and inference from AERONET retrievals.'

p9, L255-259. I think another important reason to not reduce VOC emissions is that this would impact the O3 chemistry in an unrealistic fashion. At least, unless you allow your IVOC to undergo gas-phase reactions, which I don't think is the case.

Author's response: For this work we assume reactions of IVOCs with OH regenerate the OH radical and so have no direct effect on oxidant chemistry. However, because the products of IVOC oxidation contribute to particle volume, there are slight indirect impacts on ozone via semivolatile partitioning. Figure S3 shows that from Case 1 to Case 4, the predicted $O_3$, NO and $NO_2$ diurnal patterns does not change significantly.

Changes in the manuscript: p11, L306: add 'In addition, OH oxidation of IVOCs is assumed to regenerate OH radicals and thus have minimal impact on the oxidant budget and the production of $O_3$.'

SI: add Figure S3 (as shown above)

p11, L308. Language - emission inventories cannot be "correctly modelled".

Author's response: Thanks for pointing out. We wanted to say that the inventory we used is very similar to another fuel-based inventory for CO emissions in Kim et al. (2016).

Changes in the manuscript: p13, L377: change to 'The mobile-source CO emission inventory used here (EMFAC) agrees with another fuel-based CO inventory (Kim et al., 2016), both of which reproduce the observed weekly patterns.'

p12. I don't think equations 2 and 3 are really needed, or should be moved to SI, since these are standard statistical metrics. If included (e.g. in Supplementary), then you have to explain 'Cov', 'Var', and N. You should in case explain though if you are taking statistics over hourly values. (The equations for FB and FE on page 15 could also be moved to SI.

Author's response: Yes, this is what we did. We will move the equations to SI, and add explanations.

Changes in the manuscript: We moved the equations 2 and 3 to SI, and add explanations to the terms. We also moved the equations of FB and FE to SI.

p12, L363. OOA is not really "measured" by AMS; it is estimated through the somewhat subjective procedures of PMF.

Author's response: We will change it as 'AMS-observed'.

Changes in the manuscript: change 'AMS-measured' to 'AMS-observed'.

p13, L389. Why 'However,'?

Author's response: Thanks for pointing out. We will remove it.

Changes in the manuscript: remove 'However'.

p14. With regard to these evaluations, I also wonder why biomass-burning, or COA are not addressed in more detail?

Author's response: In this paper, we focused on the SOA formation from mobile sources IVOCs. We expect to continue work on these sources in our following works.

p27, Table 2. This table gives the MW of the compounds. I am just curious - are the authors using these in their application of Raoult's law, or do they use the Donahue et al 2006 (Supplemental section) idea that one can use mass fraction rather than mole-fraction in the simulations?

Author's response: We use the MWs for organic compounds for several purposes. First, the IVOC vapors and their (larger) products are emitted, transported, and reacted in CMAQ in molar units. Thus, the mass-based SOA yields need to be adapted to convert between moles of reactant and moles of product. Second, the MW of gas-phase compounds is important for calculating the likelihood of surface uptake (deposition) to plants, water, etc.

Although they are not listed in table 2, we do use the MW of the SOA products to apply Raoult's law with mole-fractions rather than mass fractions as suggested in Donahue et al. (2006). Our view is that the assumption of rough equivalence between mass-fraction and mole-fraction approaches works well for single-precursor laboratory experiments, for which most partitioning products are of similar molecular weight. This assumption is less applicable for models like CMAQ which are combining products from lots of SOA precursors together in the same phase.

Changes in the manuscript: p7, L187: add 'The IVOC MWs are used to convert mass-based SOA yields to molar units and calculate parameters needed to simulate dry deposition processes.'

p27, Table 2. Please be explicit that yields are mass-based.

Author's response: Yes, we will add this information.

Changes in the manuscript: Table 2: change 'SOA yield' to 'mass-based SOA yield'

p29, Table 4. The term traditional is starting to lose its meaning. Do you mean inventory or officially reported POA?

Author's response: We mean inventory POA.

Changes in the manuscript: Table 4: change 'traditional' to 'inventory'.

p31, Figure 1. Give units.

Author's response: We will change volatility to volatility (log C*, μg / m$^3$).

Changes in the manuscript: p31: change volatility to volatility (log C*, μg / m$^3$).

Figs 2, 3, 6 and S1 had some formatting issues on my printer. Use a different graphics format if possible (.eps, .png?)

Author's response: We will change it to .png format.

**References**

Dunmore et al., Diesel-related hydrocarbons can dominate gas phase reactive carbon in megacities, ACP, www.atmos-chem-phys.net/15/9983/2015/

Hodzic, A. et al, A model evaluation of coarse-mode nitrate heterogeneous formation on dust particles Atmos. Environ., 2006, 40, 4158-4171

Author's response: These are very related literatures, we will add them accordingly.

**Reviewer 2**

**Comments:**

1. Line 129: The constraint to limit the SOA model to fewer than 79 additional species seems artificial. While it is true that regulatory applications of CMAQ may not want to implement calculations with large numbers of species, exploratory scientific applications of CMAQ could easily add this number of species to the SOA model. Several examples have been published in the literature (see for example papers by Ying et al. using CMAQ).

Author's response: In a paper by Ying and Li (2011), a modified chemical mechanism with 4642 species and 13,566 reactions was incorporated into the 3D CMAQ model (CMAQ-MCM) and applied to model a three-week high ozone episode in Southeast Texas. We admit that this near-explicit approach is powerful for VOCs. However, our aim is to develop a model for IVOC SOA production that can be used in off-the-shelf regulatory and routine chemical transport modeling applications. These could involve, for example, studies that will perform a large number of ensemble runs or focus on a part of the air quality system that depends indirectly on accurate PM estimates, but does not focus precisely on SOA, and thus does not want to spend precious computing resources there. From the IVOC measurement perspective, lumping similar IVOCs based on their volatility and functionality is also more interpretable and compatible to most instruments.

Changes in the manuscript: Line 133: add 'Our aim is to develop a model for IVOC SOA production that can be used in off-the-shelf regulatory and routine chemical transport modelling applications. For other applications, a more-explicit approach with multiple thousands of species may be more powerful for modelling reaction pathways (Ying and Li, 2011). From the IVOC measurement perspective, lumping similar IVOCs together based on their volatility and functionality is also more interpretable and compatible to data provide by most instruments.'

2. Line 157, eq 1: The SOA mode formulation used in the current study does not allow for fragmentation. It may not be possible to fit the coefficients in the model to adequately explain SOA formation under the full range of atmospheric aging at different VOC/NOx ratios with this limitation. Some discussion of this limitation should be included in this section that introduces the simplified SOA model.

Author's response:

We formulate the IVOC-SOA formation in two steps: (1) first generation reaction in Eq. 1 and (2) subsequent aging and fragmentation model as specified in CMAQ 5.2. The second VBS model does allow for functionalization and fragmentation of SVOCs (stated at section 3.4 and discussed in detail in Murphy et al. (2017)). We agree with the reviewer that it may not be possible to fit the coefficients in the model to *perfectly* explain SOA formation under the full range of atmospheric aging as pollutants are transported downwind of sources, even with fragmentation processes nominally accommodated. This is an area of ongoing work that has proven exceedingly difficult to constrain, mainly because the exceedingly long timescales relevant for the atmosphere are difficult to access in controlled laboratory experiments. Further, it is difficult to understand how applicable controlled lab experiments are to the aging of atmospheric pollutants that exposed to a number of changing environmental conditions and co-reactants during a multi-day trajectory.

Changes in the manuscript: Line 128: add 'first generation' and 'under high-$NO_x$ conditions'.

Line 312: add 'SOA is produced from IVOC oxidation using the parameterization described in section 2. The SOA mass is determined by CMAQ based on the gas-particle partitioning of the SVOC products created from IVOC oxidation. CMAQ v5.3 calculates partitioning assuming thermodynamic equilibrium and that all organics form a single pseudo-ideal solution. The SVOC products also undergo multigenerational aging following the approach of Murphy et al. (2017) (see section 3.4).'

3. Line 194: change "and" to "by"?

Author's response: In response to reviewer 1's comment, we changed 'remade' to 'identical to'.

4. Line 213: using diesel POA as a surrogate for all other combustion sources besides mobile, cooking, and biomass seems like a bit of a simplification. It may not matter much for the overall SOA analysis, but is this really appropriate for sources like aircraft? Or structure fires? Or natural gas combustion? Using diesel POA volatility for these sources could significantly bias the results around some localized sources.

Author's response: Yes, comparing to other sources that we don't have the profiles, such as structure fires and natural gas combustion, the surrogate (diesel POA) could either be more volatile and subject to more oxidation / aging, or less volatile. But such sources only emit 16.4% POA in LA basin during the modelled period, therefore the potential bias are small.

Changes in the manuscript: Line 257: add 'According to our emission inventory, mobile, wood-burning and cooking sources combined emit more than 80% of total POA in LA region during the modelled period, where other combustion sources only emit 16.4% of the POA. We acknowledge that the diesel POA surrogate is modestly more volatile than biomass burning POA profiles. Thus, using diesel POA volatility as the surrogate for other combustion sources will possibly increase the regional SOA formation compared to if a different profile was used, but the potential bias is small.'

5. Line 228: The authors go to great trouble to estimate the amount of additional POA that was not measured during emissions testing due to low concentrations in sampling equipment, and then describe this material as semi-volatile using POA volatility distributions (line 208). It isn't clear that this gives a different (better) answer than just leaving the original POA emissions at their nominal values and describing this material as essentially non-volatile. It would be instructive to other readers if the authors could quantify these two treatments of POA (or refer to previous publications where this has been done and summarize the results).

Author's response: The two treatments of POA emissions (non-volatile and semi-volatile) can result in similar predictions of POA concentration itself, given that the additional evaporated SVOCs may not condense again. But we add these SVOC vapors to address its semi-volatile nature (i.e. sensitivity to temperature and ambient dilution) and to best estimate the potential local and/or regional SOA formation from these SVOCs because they are subject to subsequent oxidation and aging. Robinson et al. (2007) compared the OA prediction differences of the two approaches using PMCAMx, and the results show substantial improvement in SOA prediction and urban / regional OA ratio. Murphy et al. (2017) also showed the model with representation of semivolatile POA improves predictions of hourly OA observations over the traditional nonvolatile model at multiple sites.

Changes in the manuscript: L283: add 'We add these SVOC vapors to address the bias in emissions measurements and to best estimate the potential local / regional SOA formation from mobile source SVOCs.'

6. Line 250: off-road gasoline does not account for a majority of emissions, but shouldn't off-road gasoline engines (non-catalyst) have their own unique profiles?

Author's response: We compared the on- and off-road gasoline emissions in Zhao et al. (2016) and Lu et al. (2018), and results shows that their IVOC fraction and volatility distribution are similar, both around 4% of total organic emissions. Therefore, the SOA formation should be similar enough. We used different emission profiles for VOC emissions.

7: Line 251: off-road diesel is a major source of emissions. It definitely seems like a stretch to use the on-road diesel profile to describe off-road diesel emissions. The uncertainty introduced by this issue should be analyzed in the paper.

Author's response: Yes, only limited data is available for off-road diesel (Qi et al., 2019), and it suggests the emissions are similar to on-road diesel vehicles. We assumed the consistency of IVOC emissions between on- and off-road non-DPF diesel vehicles. But we will add a sentence to address this uncertainty.

Changes in the manuscript: Line 271: add 'Although only limited data are available for off-road diesel engine emissions (Qi et al., 2019), it suggests the emissions are similar to on-road diesel vehicles. Therefore, we use the IVOC emission profile for non-DPF vehicle for all diesel sources.'

8. Line 262: Fragmentation and functionalization are accounted for, but in a biased manner. All of the SVOC and IVOC emitted in the current study will eventually make SOA due to the absence of fragmentation in the mechanism. The formation rates are tuned to account for the net effects over some pre-defined range of aging, but this simplistic model cannot capture the behavior correctly over the full lifetime. It is beyond the reasonable scope to change the simple model in the current paper, but the authors should properly describe its limitations.

Author's response: We agree with the reviewer that the parameters we choose to simulate fragmentation and functionalization are simplified compared to the full complexity of the organic gas-phase system. Our approach does allow for some fraction of SVOCs and IVOCs to form higher volatility compounds which will not partition to the particle phase. Moreover, the relative strength of this pathway increases in the model with O:C ratio of the surrogate species. Thus there is representation of the effects the author is concerned about, but the magnitude is uncertain. The aging model here was constrained to a full 2D-VBS model at 3 days of aging as discussed in Murphy et al. (2017), and the 2D-VBS model is itself a simplification of the complex kinetics for atmospheric aging. Future versions of chemical transport models like CMAQ must account for the long-term chemistry of these systems in order to properly understand the fate of particulate carbon.

Changes in the manuscript: L317: add section 3.4 Multi-generational aging and gas-particle partitioning

'The semivolatile POA emissions and semivolatile products formed from oxidation of SOA precursors undergo multigenerational aging as described in Murphy et al. (2017). Figure S5 shows the schematic diagram for OH oxidation first-generation and multigenerational aging. Briefly, the approach simulates the reaction of L/S/IVOC vapors with hydroxyl radical and distributes the product mass to a second set of five vapor-particle pairs of species at moderate O:C values. The stoichiometric ratios used to distribute the product mass were derived to match the SOA enhancement predicted by a full 2D-VBS simulation of the functionalization and fragmentation of SVOCs during three days of atmospheric oxidation. This model, unlike that of Koo et al. (2014), does transfer some of the aged products to higher volatility bins, and thus

reduces SOA over multiple generations of OH reaction. The probability for fragmentation increases as a function of O:C in agreement with theory (Donahue et al., 2011). Although the competing effects of fragmentation and functionalization at long timescales are represented in this model, the simplified framework is likely limited when trying to capture the full complexity of multigenerational aging. For this work, no changes were made to the chemical properties (e.g. carbon number, O:C, etc.) or reaction stoichiometry of the multigenerational aging mechanism of Murphy et al. (2017). Because IVOC products likely have lower carbon numbers than products of primary SVOC oxidation, our approach may represent an upper bound on the potential for IVOC SOA aging to further enhance particle mass downwind of sources.'

9. Line 312: EMFAC emissions factors are cited as a source of uncertainty, but line 199 states that mobile on-road and non-road emissions are calculated by MOVES 2014a. Which is it?

Author's response: Thank you for pointing this out. MOVEs used emission factors from EMFAC to build the emission inventory for CMAQ simulations. To be consistent, we calculate the fraction using EMFAC, and the LEV-2 vehicles account for 8.5% of total gasoline TOG emissions.

Changes in the manuscript: L386: change 'MOVES 2014a' to 'EMFAC', change '9%' to '8.5%'.

10. Line 325: The paper should also acknowledge that unknown chemical reactions leading to the formation and reaction of IVOCs could play a role in model error.

Author's response: Yes, we will include this uncertainty in the discussion.

Changes in the manuscript: L151: add 'In this study, we only account for IVOC-OH reactions because mobile-source IVOCs are mostly alkanes or aromatics, which will react slower with $O_3$. $NO_3$ oxidation can be important in night-time SOA formation (Fry et al., 2014; Hoyle et al., 2011), and these will be important to consider in the future, but experimental studies on SOA formation from anthropogenic IVOC reactions with $NO_3$ radical are limited at this time.'

11. Figure 6 lower right panel: each of these sites has a different representative atmospheric aging time. The fact that some over-predict and some under-predict as the emissions are scaled up and down may reflect the fact that the functionalization / fragmentation processes have been combined into a single lumped parameter that works at one time scale but not at others. This possibility should be discussed in the results and model formulation sections.

Author's response: Yes, we agree with the reviewer. It is very possible that the simplifications in the aging model lead to discrepancies in performance among sites. As stated in the response to Comment 8 above, the approach used here is more complex than a single lumped parameter and we believe the additional data we have included helps to capture more of the long-term fragmentation behavior than the model of Koo et al. (2014). It is critical in the future to constrain aging models with ambient observations of downwind aging if we are to reduce this uncertainty further.

Changes in the manuscript: L545: add 'This could be due to the different meteorological conditions, or source variations, and/or inaccuracies in the multigenerational aging model. More research is needed to better understand the competition between functionalization and fragmentation of organic gases at long atmospheric timescales.'